# 2010-2015 North American methane emissions, sectoral contributions, and trends: a high-resolution inversion of GOSAT satellite observations of atmospheric methane

Joannes D. Maasakkers[1,2], Daniel J. Jacob[1], Melissa P. Sulprizio[1], Tia R. Scarpelli[1], Hannah Nesser[1], Jianxiong Sheng[1,3], Yuzhong Zhang[1,4,5,6], Xiao Lu[1], A. Anthony Bloom[7], Kevin W. Bowman[7,8], John R. Worden[7], and Robert J. Parker[9,10]

[1]Harvard University, Cambridge, Massachusetts 02138, United States
[2]SRON Netherlands Institute for Space Research, Utrecht, The Netherlands
[3]Massachusetts Institute of Technology, Cambridge, MA, United States
[4]Environmental Defense Fund, Washington, DC, USA
[5]School of Engineering, Westlake University, Hangzhou, Zhejiang Province, China
[6]Institute of Advanced Technology, Westlake Institute for Advanced Study, Hangzhou, Zhejiang Province, China
[7]Jet Propulsion Laboratory, California Institute of Technology, Pasadena, CA, USA
[8]Joint Institute for Regional Earth System Science and Engineering, University of California, Los Angeles, CA, USA
[9]Earth Observation Science, School of Physics and Astronomy, University of Leicester, Leicester, UK
[10]NERC National Centre for Earth Observation, Leicester, UK

**Correspondence:** J.D. Maasakkers (j.d.maasakkers@sron.nl)

**Abstract.** We use 2010-2015 GOSAT satellite observations of atmospheric methane columns over North America in a high-resolution inversion of methane emissions, including contributions from different sectors and their trends over the period. The inversion involves analytical solution to the Bayesian optimization problem for a Gaussian mixture model (GMM) of the emission field with up to $0.5° \times 0.625°$ resolution in concentrated source regions. Analytical solution provides a closed-form characterization of the information content from the inversion and facilitates the construction of a large ensemble of solutions exploring the effect of different uncertainties and assumptions in the inverse analysis. Prior estimates for the inversion include a gridded version of the EPA Inventory of U.S. Greenhouse Gas Emissions and Sinks (GHGI) and the WetCHARTS model ensemble for wetlands. Our best estimate for mean 2010-2015 US anthropogenic emissions is 30.6 (range: 29.4-31.3) Tg $a^{-1}$, slightly higher than the gridded EPA inventory (28.7 (26.4-36.2) Tg $a^{-1}$). The main discrepancy is for the oil and gas production sectors where we find higher emissions than the GHGI by 35% and 22% respectively. The most recent version of the EPA GHGI revises downward its estimate of emissions from oil production and we find that these are a factor 2 lower than our estimate. Our best estimate of US wetland emissions is 10.2 (5.6-11.1) Tg $a^{-1}$, on the low end of the prior WetCHARTS inventory uncertainty range (14.2 (3.3-32.4) Tg $a^{-1}$) and calling for better understanding of these emissions. We find an increasing trend in US anthropogenic emissions over 2010-2015 of 0.4% $a^{-1}$, lower than previous GOSAT-based estimates but opposite to the decrease reported by the EPA GHGI. Most of this increase appears driven by unconventional oil/gas production in the eastern US. We also find that oil/gas production emissions in Mexico are higher than in the nationally reported inventory, though there

is evidence for a 2010-2015 decrease in emissions from offshore oil production.

## 1 Introduction

Methane is the second most important greenhouse gas in terms of radiative forcing (Stocker et al., 2013). Global methane concentrations have increased by a factor 2.5 compared to preindustrial times Hartmann et al. (2013) and by 7.1 ppb $^{-1}$ since 2007 (with the rate peaking above 10 ppb a$^{-1}$ in 2014 and 2015) after a period of stability in the early 2000s (https://www.esrl.noaa.gov/gmd/ccgg/trends_ch4/, last access: 20 December 2020). Major emission source sectors include wetlands (the main natural source), livestock, the fossil fuel industry, and waste management (Kirschke et al., 2013; Saunois et al., 2020). Individual countries report their anthropogenic emissions to the United Nations Framework Convention on Climate Change (UNFCCC) using methods prescribed by The Intergovernmental Panel on Climate Change (IPCC) (United Nations, 1992; IPCC, 2006). The reports use "bottom-up" methods, where activity data (e.g., heads of cattle) are combined with emission factors (e.g., emission per head of cattle) to estimate total emissions. US emissions are calculated and reported in this manner by the Environmental Protection Agency (EPA) in its annual Inventory of U.S. Greenhouse Gas Emissions and Sinks (Greenhouse Gas Inventory, GHGI) (EPA, 2020). Measurements of atmospheric methane, including from satellites, can be used through inverse modeling to provide an evaluation of these emission estimates (Streets et al., 2013; Jacob et al., 2016). Here, we evaluate 2010-2015 North American emissions by inversion of data from the Greenhouse Gases Observing Satellite (GOSAT), which measures methane concentrations at high precision by solar backscatter in the shortwave infrared (SWIR) (Butz et al., 2011; Buchwitz et al., 2015; Kuze et al., 2016). We take the gridded version of the EPA GHGI (Maasakkers et al., 2016) as prior estimate for the inversion, enabling us to use the inversion results to evaluate the GHGI and guide improvements in its representation of emission processes.

Several inverse studies using observations of atmospheric methane have found higher US emissions than reported in bottom-up inventories. Miller et al. (2013) estimated methane emissions 50% higher than the EPA GHGI based on 2007-2008 surface and aircraft observations. They attributed this difference to fossil fuel extraction. Wecht et al. (2014) estimated 40% higher livestock emissions using 2004 data from the SCIAMACHY satellite instrument (Frankenberg et al., 2006). Turner et al. (2015) found anthropogenic emissions to be 50% higher than the EPA GHGI by inversion of 2009-2011 GOSAT data, attributing the difference to a combination of oil/gas and livestock emissions. Janardanan et al. (2017) found 28% higher anthropogenic emissions over North America based on 2009-2012 GOSAT data. All these studies used the global gridded EDGAR inventory (European Commission, 2011; Olivier and Janssens-Maenhout, 2012; Janssens-Maenhout et al., 2019) as prior estimate, but the EDGAR spatial distributions have large errors that affect inversion results and their interpretation (Maasakkers et al., 2016). Alvarez et al. (2018) used surface and aircraft data in oil/gas fields to find emissions from oil/gas production 60% higher than in the EPA GHGI.

There has also been substantial debate as to the contribution of North American emissions to the rising global methane trend since 2007. Hausmann et al. (2016) proposed an increase in US gas production as explanation for the 2007-2014 methane and ethane concentration trends at the Zugspitze mountain site in southern Germany. Turner et al. (2016) found a 2.5% a$^{-1}$ increase in US emissions for 2002-2014 on the basis of GOSAT and surface methane data. By contrast, Bruhwiler et al. (2017) found from an ensemble of inversions using surface and satellite observations that North American emissions had been flat for 2000-2012 and that without an inverse model short-term trends can appear to be present in the GOSAT data because of interannual transport variability, choice of background, and the seasonal sampling bias of GOSAT. Sheng et al. (2018a) analyzed 2010-2016 GOSAT enhancements over North America and found a $2.5 \pm 1.4$ % a$^{-1}$ increase over the US driven by oil/gas production and livestock emissions in the Midwest. They found no significant trend over Canada (but large year-to-year variation driven by wetlands) and a 0.8-1.7 % a$^{-1}$ decrease in Mexican emissions driven by a decrease in livestock. Using 2006-2015 surface and aircraft observations over the US and Canada, Lan et al. (2019) found a $0.7 \pm 0.3$ % a$^{-1}$ increase in total US emissions and a $3.4 \pm 1.4$ % a$^{-1}$ increase in oil/gas emissions based on stations in Oklahoma, North Dakota, and Texas. They also reported an increase in the ethane:methane emission ratio which could lead to an overestimate of the oil/gas methane trend as inferred from the ethane trend.

Our inverse analysis of the 2010-2015 GOSAT data over North America optimizes both mean emissions and their long-term trends at high resolution (up to 50 km). The inversion uses dynamic boundary conditions from a consistent global inversion of the 2010-2015 GOSAT data previously reported by Maasakkers et al. (2019). We use an analytical solution to the Bayesian inverse optimization problem (Jacob et al., 2016), which provides closed-form characterization of the information content of the solution, and also enables us to perform a range of sensitivity inversions (inversion ensemble) at no added computational cost. We relate the results from our inversion ensemble to the EPA GHGI emissions in order to inform knowledge of US emissions, their trends, and the contributions from different sectors.

## 2 Data and methods

We perform a continental-scale inversion of 2010-2015 GOSAT methane data from the University of Leicester proxy CH$_4$ retrieval (Parker et al., 2015; ESA CCI GHG project team , 2018). We use the individual GOSAT observations to optimize a state vector of mean methane emissions and linear emission trends trends at spatial resolution of up to $0.5° \times 0.625°$ ($\approx 50 \times 50$ km$^2$). The forward model for the inversion is the GEOS-Chem chemical transport model (www.geos-chem.org) applied in nested mode over North America with a spatial resolution of $0.5° \times 0.625°$. The main results presented here will be from a base inversion with specifications given below. In addition to this base inversion, we conducted an ensemble of 9 inversions in which we explored the sensitivity of the results to different assumptions. Specifications for these sensitivity inversions are given below and summarized in Section 2.5. The posterior error covariance matrix from the inversion underestimates the actual uncertainty in the results because of the assumption of fully random observational errors (Jacob et al., 2016). Therefore we use

the range of results from the inversion ensemble as a better measure of uncertainty (Heald et al., 2004).

## 2.1 GOSAT observations

The GOSAT satellite has been observing dry column methane mixing ratios in the SWIR using the TANSO-FTS instrument
since April 2009 (Butz et al., 2011). GOSAT in its default mode observes circular pixels of 10 km diameter at 13:00 local time, separated by $\sim$250 km along-track and cross-track, repeating observation on the same track every 3 days. Additional locations are be observed using a target mode. GOSAT methane retrievals have a 24% success rate, limited mainly by cloud cover. Observations have a precision of 13 ppb and relative bias of 2 ppb compared to the Total Carbon Column Observing Network (Buchwitz et al., 2015). There has been no significant spectral degradation of the observations over time (Kuze et al., 2016). Figure 1 shows the 156,110 retrievals over land used to optimize emissions in this study (Parker et al., 2015). Each retrieval comes with an estimated retrieval error (11 ppb on average). We use observations over land from January 2010 to December 2015, excluding data above 60N° for which model errors are large (Maasakkers et al., 2019). Most observations (95,365) are over the Contiguous United States (CONUS). The data are spatially sparse but this reflects the observing strategy of repeated measurements at the same locations in the default mode. Thus most observation locations in Figure 1 have a large number of data points to inform temporal variability and trends (Sheng et al., 2018a).

## 2.2 Prior estimates

Table 1 summarizes our prior emissions estimates and Figure 2 shows their spatial distributions for the major sectors. For all US anthropogenic emissions including offshore as reported to the UNFCCC, we use the spatially disaggregated (gridded) version of the EPA GHGI (EPA, 2016) for 2012 from Maasakkers et al. (2016), with improved spatial allocation of emissions and detailed separation of source sectors compared to EDGAR. For oil/gas emissions in Mexico and Canada including offshore, we use gridded versions of the Instituto Mexicano del Petróleo (IMP) inventory for 2010 (IMP, 2012) and the ICF International inventory for 2013 (ICF, 2015), respectively, as reported by Sheng et al. (2017). IMP (2012) oil/gas emissions for Mexico match the values reported by Mexico to UNFCCC, while ICF (2015) oil/gas emissions for Canada are 43% higher than the value reported by Canada (UNFCCC, 2019). The ICF inventory was used as the basis for the Sheng et al. (2017) gridded emission inventory as it provides a detailed breakdown of sources using methodology similar to the EPA GHGI. For other anthropogenic emissions in Canada/Mexico and other countries in the domain, we use the EDGAR v4.3.2 global emission inventory for 2012 (edgar.jrc.ec.europa.eu [2017]). We separate the general 'fuel exploitation' sector reported by EDGAR v4.3.2 into oil/gas and coal components by using additional information embedded in the inventory (Greet Maenhout, personal communication). This allows us to use EDGAR coal emissions for Canada and Mexico. The emissions as given by EDGAR v4.3.2 are aseasonal. For manure management and rice cultivation we apply seasonality as prescribed by Maasakkers et al. (2016) and Zhang et al. (2016) respectively. Other anthropogenic emissions remain aseasonal.

Natural emissions are dominated by wetlands for which we use mean monthly emissions from the WetCHARTS v1.0 extended ensemble with $0.5° \times 0.5°$ spatial resolution (Bloom et al., 2017). The ensemble parameters consist of: three global scaling factors (global emissions of 124.5, 166, or 207.5 Tg a$^{-1}$); three temperature q10 dependencies (1,2, or 3); and two landcover databases that are combined with precipitation data to estimate wetland extent (GLWD from Lehner and Döll (2004) or GlobCover from Bontemps et al. (2011)). Total wetland emissions vary month to month and interannually as driven by temperature and inundation extent (Bloom et al., 2017). Because the WetCHARTs ensemble exhibits considerable but uncertain year-to-year variability, we also perform a sensitivity inversion without prior interannual variability in wetland and other emissions. Daily open fire emissions are from the Quick Fire Emissions Dataset (QFED) (Darmenov and da Silva, 2013) and termite emissions are from Fung et al. (1991) with a global total of 12 Tg a$^{-1}$. We use geological seepage emissions compiled from literature on both point sources (Etiope, 2015; Kvenvolden and Rogers, 2005) and areal seepage (Kvenvolden and Rogers, 2005; Etiope and Klusman, 2010) as described in Maasakkers et al. (2019) with a global total of 5 Tg a$^{-1}$, under the 5.4 Tg a$^{-1}$ maximum proposed for pre-industrial times by Hmiel et al. (2020) based on ice core measurements.

## 2.3 Forward model

We use the nested version of the GEOS-Chem chemical transport model v11-01 at $0.5° \times 0.625°$ resolution over North America as forward model for the inversion. Earlier versions of this model for methane were described by Wecht et al. (2014) and Turner et al. (2015). The model is driven with MERRA-2 meteorological fields (Bosilovich et al., 2016) from the NASA Global Modeling and Assimilation Office (GMAO). Methane loss from reaction with OH and Cl radicals, soil uptake, and stratospheric oxidation are described in Maasakkers et al. (2019). The simulation is initialized in January 2009 with concentration fields from Turner et al. (2015). Three-hourly boundary conditions at the edges of the nested domain are from the $4° \times 5°$ posterior model simulation of Maasakkers et al. (2019), which provides an unbiased fit to the global GOSAT data. That posterior simulation includes some information from GOSAT data over the North America domain, which were used (along with the more abundant data outside that domain) in the global inversion; but the main consideration here is to avoid bias in boundary conditions that would otherwise affect the North American inversion. Methane chemical and soil sinks are not optimized in our inversion because they are very slow compared to the time scale for ventilation of the North American domain.

Following Maasakkers et al. (2019), we correct the GEOS-Chem simulation of GOSAT columns for a latitudinally and seasonally variable background bias likely caused by the extratropical stratosphere (Bader et al., 2016; Saad et al., 2016; Stanevich, 2018). The bias is common in atmospheric models and caused by excessive meridional transport in the stratosphere (Patra et al., 2011) and in particular in the seasonal polar vortices (Zhang et al., 2020b). The latitudinal correction term $\xi$ (ppb) follows a quadratic form as in Turner et al. (2015):

$$\xi = \left(4.0\theta^2 - 1.3\theta\right) \times 10^{-3} - 5 \tag{1}$$

**Table 1.** Methane emissions used as prior 2010-2015 estimates[a].

| Source (Tg a$^{-1}$) | CONUS | Canada | Mexico | Other[b] |
|---|---|---|---|---|
| Natural | 15.7 | 15.3 | 1.4 | 3.8 |
|     Wetlands | 14.2 | 14.4 | 1.0 | 3.4 |
|     Open fires | 0.5 | 0.3 | 0.2 | 0.1 |
|     Termites | 0.6 | 0.3 | 0.1 | 0.2 |
|     Geological seeps | 0.5 | 0.3 | 0.1 | <0.1 |
| | | | | |
| Anthropogenic | 28.7 | 4.5 | 5.3 | 5.1 |
|     Livestock | 9.2 | 1.0 | 2.5 | 1.9 |
|         Enteric Fermentation | 6.7 | 0.8 | 2.2 | 1.9 |
|         Manure Management | 2.5 | 0.2 | 0.3 | 0.1 |
|     Oil and Natural Gas | 9.1 | 2.4 | 1.5 | 1.2 |
|         Gas production | 4.4 | 1.2 | 0.1 | |
|         Oil production | 2.3 | 0.5 | 1.2 | |
|         Gas Transmission | 1.1 | 0.3 | <0.1 | |
|         Gas Processing | 0.9 | 0.3 | 0.1 | |
|         Gas Distribution | 0.5 | <0.1 | <0.1 | |
|     Landfills | 5.8 | 0.7 | 0.4 | 0.6 |
|     Coal Mining | 2.9 | 0.1 | <0.1 | <0.1 |
|     Wastewater | 0.7 | 0.2 | 0.7 | 0.7 |
|     Rice Cultivation | 0.5 | <0.1 | <0.1 | 0.2 |
|     Other Anthropogenic[c] | 0.5 | 0.1 | 0.2 | 0.4 |
| | | | | |
| Total Source | 44.5 | 19.8 | 6.7 | 8.9 |

[a] CONUS anthropogenic emissions are from the EPA GHGI for 2012 as spatially disaggregated by Maasakkers et al. (2016). Oil/gas emissions from Canada (2013) and Mexico (2010) are from ICF (2015) and IMP (2012), respectively, spatially disaggregated by Sheng et al. (2017). Other anthropogenic emissions are from EDGAR v4.3.2 for 2012 (edgar.jrc.ec.europa.eu [2017]). Wetlands and open fire emissions are mean values for 2010-2015 from the WetCHARTS ensemble (Bloom et al., 2017) and QFED (Darmenov and da Silva, 2013); Termite emissions are from Fung et al. (1991). Seepage emissions are as described in Maasakkers et al. (2019). The soil sink is 3.6 Tg a$^{-1}$ for the inversion domain (Fung et al., 1991) and is not optimized in the inversion. All values in the table are rounded to one decimal.

[b] within the inversion domain shown in Figure 1 (10-70N$^\circ$, 140-40W$^\circ$) containing parts of Central and South America.

[c] including fossil fuel combustion, industrial processes, agricultural field burning, and composting.

with $\theta$ the latitude in degrees. The seasonal bias is corrected over rolling 8° latitudinal bands. A sensitivity inversion without the seasonal bias correction is performed as part of the inversion ensemble.

## 2.4    State vector for the inversion and error covariances

Although we could technically carry out the inversion of the GOSAT data at the $0.5° \times 0.625°$ resolution of the GEOS-Chem simulation, the data do not have sufficient information to constrain emissions on that grid and doing so would incur large smoothing error (Wecht et al., 2014). We use instead a 600-element Gaussian mixture model (GMM) as described by Turner and Jacob (2015) to optimally define the emission patterns that can be usefully constrained by the inversion. Each of the 600 Gaussian functions in the GMM is defined by an emission amplitude, mean location, and spread (standard deviation). These

parameters are optimized using a similarity vector on the $0.5° \times 0.625°$ grid that takes into account latitude, longitude, and the prior patterns of different source sectors. The state vector $\mathbf{x}$ for the inversion with dimension $n = 2 \times 600$ consists of scaling factors adjusting the amplitudes of the Gaussians in the GMM and their 2010-2015 linear trends. This approach allows for effective aggregation of regions with weak or homogeneous emissions while preserving high resolution for concentrated emissions. Each $0.5° \times 0.625°$ grid cell is represented by a unique combination of the Gaussians, so that the optimization of $\mathbf{x}$

can be mapped to the $0.5° \times 0.625°$ grid. For more details see Turner and Jacob (2015).

Prior emission error variances are defined for each Gaussian on the basis of its spatial distribution and the contributions from different sectors. Emission errors for individual anthropogenic sectors are estimated using the error curves from Maasakkers et al. (2016). The error standard deviation $\sigma$ for a given source sector and Gaussian is given by:

$$\sigma = (\alpha_0 \exp(-k_\alpha(L - L_0)) + \alpha_N) E \qquad (2)$$

Where $\alpha_0$, $k_\alpha$, and $\alpha_N$ are source sector specific error coefficients from Maasakkers et al. (2016), $L$ is the effective spatial resolution (length scale) of the Gaussian defined by the number of $0.5° \times 0.625°$ grid cells it represents, $L_0 = 0.1°$ is the native resolution of the prior inventory, and $E$ is the sum of emissions from the source sector within the Gaussian (sum of emissions from $0.5° \times 0.625°$ grid cells weighted by their contributions to the Gaussian). Maasakkers et al. (2016) also include a dis-

placement error related to uncertainty in source location but this error is negligible at our resolution. For wetland emissions, we use the standard deviation in monthly estimates of the 18 WetCHARTS v1.0 extended ensemble members averaged over the Gaussian; the resulting error standard deviation is 78% on average. For the other natural emissions we assume 100% error at the $0.5° \times 0.625°$ model resolution.

The error variances for all sectors contributing to a given Gaussian are added in quadrature to obtain the corresponding diagonal element of the prior error covariance matrix $\mathbf{S_A}$. Error variances for a given Gaussian are capped at 50% in the base inversion and we also perform a sensitivity inversion without this cap. The mean relative error standard deviation is 37% in the

base inversion. The 50% cap mainly affects Gaussians dominated by wetland emissions. For the 2010-2015 emission trends associated with each Gaussian, the prior estimate is set to zero and the prior error standard deviation is a 5% change per year, in line with uncertainties in trend estimates for North America (Turner et al., 2016; Bruhwiler et al., 2017; Sheng et al., 2018a; Lan et al., 2019). We also perform sensitivity inversions with changes of 2.5% and 10% per year as prior error standard devia-

5 tion. Off-diagonal elements of $\mathbf{S_A}$ are assumed to be zero because Maasakkers et al. (2016) found no spatial error correlation for the gridded EPA inventory; this may be an underestimate for wetland emissions (Bloom et al., 2017).

Our calculation of $\mathbf{S_A}$ leads to different error variances for each grid cell. To assess the impact of that choice, we also perform a sensitivity inversion using the mean error variance for all the Gaussians. The base inversion assumes normal errors,

but we also perform a sensitivity inversion assuming log-normal emission errors following the Levenberg-Marquardt method as described in Maasakkers et al. (2019).

We use the residual error method (Heald et al., 2004) to construct the diagonal of the observational error covariance matrix $\mathbf{S_O}$. The mean 2010-2015 difference between GOSAT and the prior model (before seasonal correction) for each $0.5° \times 0.625°$

grid cell is assumed to be due to errors in emissions, to be corrected by the inversion. After subtracting this mean difference, the residual standard deviation is taken as estimate of the observational error standard deviation including contributions from instrument, representation, and forward model errors. If this estimate is less than the reported instrument error standard deviation (Parker et al., 2015), we use the latter instead (17% of observations). If it is less than 10 ppb we reset it to 10 ppb (6% of observations). The resulting average observational error standard deviation is 14 ppb. Off-diagonal terms of $\mathbf{S_O}$ are assumed

to be zero for lack of better information, but in fact some transport error correlation would be expected in the forward model. We account for this error correlation with a regularization term $\gamma$ in the inversion (Section 2.5).

## 2.5    Inversion procedure

We perform an analytical inversion minimizing the Bayesian cost function $J(\mathbf{x})$ assuming normal errors (Rodgers, 2000):

$$J(\mathbf{x}) = (\mathbf{x} - \mathbf{x_A})^T \mathbf{S_A}^{-1} (\mathbf{x} - \mathbf{x_A}) + \gamma (\mathbf{y} - F(\mathbf{x}))^T \mathbf{S_O}^{-1} (\mathbf{y} - F(\mathbf{x})) \tag{3}$$

where $\mathbf{x}$ is the state vector to be optimized, consisting of 600 Gaussians for which we optimize both scaling factors for mean emissions and absolute linear emission trends, for a total of 1200 state vector elements; $\mathbf{x_A}$ is the prior state vector; $\mathbf{S_A}$ is the prior error covariance matrix (Section 2.4); $\mathbf{S_O}$ is the observational error covariance matrix (Section 2.4); and $\gamma$ is a regularization factor to account for the lack of non-diagonal terms in $\mathbf{S_O}$ and hence prevent overfitting. $\gamma$ plays a similar role

as the regularization parameter in Tikhonov methods (Brasseur and Jacob, 2017) and reflects our inability to precisely quantify error statistics in the Bayesian method. Here we find that $\gamma = 0.5$ provides the best balance of fitting the prior and observational terms in the cost function, following the L-curve approach of Hansen (1999). The value is higher than $\gamma = 0.05$ used in the global inversion of Maasakkers et al. (2019) at $4° \times 5°$ resolution because here we have a smaller number of observations per

state vector element. We also conduct sensitivity inversions with $\gamma = 0.1$ and $\gamma = 1$.

The GEOS-Chem forward model ($\mathbf{y} = F(\mathbf{x})$) as implemented here is strictly linear in its relationship between methane column concentrations ($\mathbf{y}$) and the state vector of emissions ($\mathbf{x}$). It can be expressed as $F(\mathbf{x}) = \mathbf{Kx} + \mathbf{c}$ where $\mathbf{K} = \partial \mathbf{y}/\partial \mathbf{x}$ is the Jacobian matrix and $\mathbf{c}$ an initialization constant. This allows the optimal posterior solution $\widehat{\mathbf{x}}$ which minimizes the cost function $J(\mathbf{x})$ to be obtained analytically as:

$$\widehat{\mathbf{x}} = \mathbf{x_A} + \mathbf{S_A}\mathbf{K}^T \left( \mathbf{K}\mathbf{S_A}\mathbf{K}^T + \frac{\mathbf{S_O}}{\gamma} \right)^{-1} (\mathbf{y} - \mathbf{Kx_A}) \tag{4}$$

with posterior error correlation matrix $\widehat{\mathbf{S}}$:

$$\widehat{\mathbf{S}} = \left( \gamma \mathbf{K}^T \mathbf{S_O}^{-1} \mathbf{K} + \mathbf{S_A}^{-1} \right)^{-1} \tag{5}$$

The information content from the inversion can then be obtained from the averaging kernel matrix ($\mathbf{A} = \partial \widehat{\mathbf{x}}/\partial \mathbf{x}$) which gives the sensitivity of the solution to the true state:

$$\mathbf{A} = \mathbf{I} - \widehat{\mathbf{S}}\mathbf{S_A}^{-1} \tag{6}$$

The trace of $\mathbf{A}$ gives the degrees of freedom for signal (DOFS), which measures the number of independent pieces of information on the state vector that can be obtained from the inversion. The diagonal elements of $\mathbf{A}$ (averaging kernel sensitivities) measure the degree to which the inversion can constrain the true values of the corresponding state vector elements (1 = perfectly, 0 = not at all). We will use these measures of information in our presentation of results.

Analytical solution to the inverse problem requires explicit construction of the Jacobian matrix. We perform this construction column by column by perturbing individually the 1200 elements of the state vector and conducting the corresponding GEOS-Chem simulations for the 2010-2015 observation record. This is readily done as a massively parallel calculation. Aside from enabling closed-form characterization of the information content from the inversion, a major advantage of the analytical solution once the Jacobian matrix has been constructed is that the sensitivity of the solution to various assumptions and choices made in the inversion approach can be immediately obtained. In addition to our base inversion, we generate in this manner an ensemble of nine sensitivity inversions introduced in the text above, and for which the ensemble of solutions gives a better measure of posterior error than can be obtained from $\widehat{\mathbf{S}}$ (Heald et al., 2004). To summarize, these sensitivity inversions include: (1) Using a prior estimate with no interannual variability (2012 values) for wetland and biomass burning emissions, and for the soil sink; (2) Not using a seasonal correction to the GOSAT - model mismatch; (3) Using emission error variances without the 50% cap; (4) Using the average emission and absolute trend error variances (37% and 2.3 Mg a$^{-2}$ km$^{-2}$, respectively) for each Gaussian; (5,6) Assuming error prior standard deviations for the 2010-2015 trend of 2.5% and 10% annual change of emissions (instead of 5%); (7) Assuming log-normal prior emission errors; (8,9) Using regularization factors for the cost

function $\gamma = 0.1$ and $\gamma = 1$ (instead of $\gamma = 0.5$).

## 3 Results and Discussion

Figure 3 shows mean prior and posterior emissions for 2010-2015, the ratio between the two, and the inversion's averaging
kernel sensitivities (Equation 6). The averaging kernel sensitivities identify regions where the GOSAT observations provide
significant information on emissions. These are regions with a high density of observations and/or high absolute uncertainties
on the prior emissions. For example, we achieve good constraints on emissions in central Canada, much of the eastern and cen-
tral US, California, and southeastern Mexico. Other regions receive little information from the observations, which explains a
lack of departure from the prior estimate.

The posterior emissions when implemented in GEOS-Chem reduce the mean squared difference with GOSAT observa-
tions by 3.5%. This overall reduction in error is small because random errors in individual observations are large and because
the background is already captured well in the prior simulation through the optimized boundary conditions. The main im-
provements are found over areas where the averaging kernels are large (Figure 3). For data with averaging kernel sensitivities
greater than 0.1, the mean squared difference is reduced by 6.1% and the correlation increases from $0.62$ to $0.64$. We inde-
pendently evaluated the posterior estimate by comparison to in situ methane concentrations from surface sites reported in the
GLOBALVIEWplus CH4 ObsPack v1.0 data product compiled by the NOAA Global Monitoring Laboratory (Cooperative
Global Atmospheric Data Integration Project, 2019). Compared to the prior simulation (Reduced major axis (RMA) slope:
$0.69$, $r^2 = 0.39$), the posterior simulation (RMA slope: $0.69$, $r^2 = 0.45$) does not degrade the comparison with these data and
improves the correlation. The spatial coefficient of determination between the time-averaged GEOS-Chem and NOAA data
increases from $r^2 = 0.58$ with the prior emissions to $r^2 = 0.81$ with the posterior emissions, representing an improvement in
our ability to fit observed patterns.

### 3.1 Mean 2010-2015 emissions

Although the inversion yields little change in total emissions for the continental domain, there are large regional changes as
shown in Figure 3. We find higher emissions over the south-central and eastern US, and lower emissions in California com-
pared to the gridded EPA inventory. The WetCHARTS inventory overestimates wetland emissions including along the Gulf
and east coasts of the US, the upper Midwest, and Canada. Emissions in eastern Mexico are higher than inferred from the
IMP (2012) inventory. The inversion also shows large relative increases from oil production off the Louisiana coast and from
wetlands/livestock in western Montana but the associated emissions are low. The large-scale correction patterns revealed by
the inversion are similar to those of the coarse ($4° \times 5°$) global inversion reported by Maasakkers et al. (2019), which used the

same prior estimates, but we have much more detail here allowed by the higher resolution of the inversion.

Figure 4 shows the attribution of the inversion results to individual source sectors for CONUS, Canada, and Mexico. This attribution was made by applying the correction factors to the sectoral emissions in each grid cell, assuming that the relative contributions of individual sectors to emissions in that grid cell is correct in the prior emission inventory (this does not assume that the total prior distribution of sectoral emissions is correct). Vertical bars show the range of results from the inversion ensemble. A narrow uncertainty range does not necessarily reflect confidence in the inversion results. For small source sectors, it may also be due to insufficient information from the observations so that the optimization is unable to depart from the prior estimate. This can be determined using the averaging kernel sensitivities, as will be done below for the US (see Table 2).

The largest decrease is for US wetland emissions, mostly contributed by the Gulf and east coasts (Figure 2). Such an over-estimate in the mean of the WetCHARTS wetland inventory ensemble was previously identified in an inversion of aircraft observations over the Southeast US (Sheng et al., 2018b). It may be related to the low organic carbon content of the soil, the difficulty of distinguishing freshwater and saltwater wetlands, uncertainties in anaerobic $CH_4:CO_2$ respiration rates, and the accounting of partial wetland land-cover areas (Holmquist et al., 2018; Lehner and Döll, 2004; Bloom et al., 2017). We also find an overestimate of wetland emissions in (mainly eastern) Canada. The large uncertainty range is driven by the inversion ensemble member without seasonal correction. Based on the root mean square error and spatial correlation, our inversion results are most consistent with the WetCHARTs ensemble members that use GlobCover wetland extent, a $q_{10} = 2$ value for the factor increase in the $CH_4$ to $CO_2$ emission ratio per 10 K temperature increase (a critical quantity for determining the sensitivity of wetland $CH_4$ production to temperature (Yvon-Durocher et al., 2014; Bloom et al., 2016)), and global scaling at the low end or middle of the range (global wetlands emission range 125-166 Tg a$^{-1}$). A value of $q_{10} = 2$ is approximately equivalent with the average $CH_4$ to $CO_2$ temperature sensitivity reported by Yvon-Durocher et al. (2014) based on meta-analyses, which indicates that anaerobic $CH_4$ respiration is substantially more sensitive to temperature relative to overall $CO_2$ respiration rates. Sheng et al. (2018b) also found their inversion results to be most consistent with GlobCover but also favored no $CH_4:CO_2$ temperature dependence ($q_{10} = 1$). Their observations were much more limited in space and time (August-September 2013).

Figure 4 also shows some significant sectoral corrections for anthropogenic emissions in the US and Mexico. Over Mexico higher livestock (+13 (5-24)%) and oil/gas emissions (+22 (-24-42)%). Uncertainty ranges in correcting individual sectors are large for Mexico because of the extensive spatial overlap between sectors (Figure 2). Most of the oil/gas correction is for coastal/offshore oil production (Figure 3). We also find 56 (31-120) % higher emissions over Mexico City, which is optimized by a single Gaussian covering five grid cells. The difference is attributed to wastewater based on the EDGAR spatial patterns. Compared to EDGAR v4.3.2 a recent gridded inventory for Mexico (Scarpelli et al., 2020) and the Mexico City Secretariat of Environment (SEDEMA, 2018) air quality emission inventory predict lower emissions from wastewater (68 versus 259 Gg a$^{-1}$ in the SEDEMA inventory) but much higher landfill emissions (222 versus 1 Gg a$^{-1}$) indicating that our higher emission

estimate may be related to landfill emissions being misallocated in EDGAR v4.3.2.

Inversion results for the US mapped on to the detailed source sectors and subsectors from the gridded EPA inventory (Maasakkers et al., 2016) are given in Table 2. The Table also includes the averaging kernel sensitivity $a_{i,i}$ from the inversion for each sector and subsector $i$, which we estimate by summing emissions from sector/subsector $i$ for all $0.5° \times 0.625°$ grid cells over the CONUS into one state vector element using a summation matrix (Calisesi et al., 2005; Maasakkers et al., 2019). The summation matrix ($\mathbf{W}$) weighs the relative contribution $w_{i,k}$ of sector/subsector $i$ to the total emission in Gaussian $k$ in the prior inventory:

$$\mathbf{A_{sub}} = \mathbf{WAW}^* \tag{7}$$

$$\widehat{\mathbf{S}}_{\mathbf{sub}} = \mathbf{W}\widehat{\mathbf{S}}\mathbf{W^T} \tag{8}$$

Here $\mathbf{W}^* = \mathbf{W}^T \left(\mathbf{WW}^T\right)^{-1}$ is the generalized pseudo-inverse of $\mathbf{W}$, and $\mathbf{A_{sub}}$ (with diagonal elements $a_{i,i}$) and $\widehat{\mathbf{S}}_{\mathbf{sub}}$ are the averaging kernel matrix and posterior error covariance matrix mapped to the different subsectors. $a_{i,i} = 1$ means that the inversion can fully constrain the national total for that emission category, independent of the prior estimate, while $a_{i,i} = 0$ means that the inversion provides no information and the estimate cannot depart from the prior. The off-diagonal elements of $\widehat{\mathbf{S}}_{\mathbf{sub}}$ measure the error correlation in the posterior solution for different subsectors, and this is important to diagnose whether we can optimize different subsectors independently. The diagonal elements of $\widehat{\mathbf{S}}_{\mathbf{sub}}$ estimate the error variance in the posterior solution for individual subsectors but that estimate is too small because it assumes that the observations are independent and identically distributed (IID condition) (Brasseur and Jacob, 2017). We prefer to estimate the error in the posterior solution from the results of the inversion ensemble, as shown in Table 2.

The averaging kernel sensitivities for individual sectors/subsectors in Table 2 vary based on the uncertainty of the prior emission estimates and the GOSAT observation density in the regions of emissions. Posterior wetland emissions ($a_{i,i} = 0.70$) are 70% informed by the observations (30% by the prior) because the prior uncertainty is large. We also calculate $a_{i,i}$ for the sum of US anthropogenic emission categories and find emissions are 53% informed by the observations, with less information for individual sectors/subsectors. Emissions from oil/gas production are particularly well informed (28-52%) because they are large and have relatively little spatial overlap with other sectors.

Our posterior estimate for the mean 2010-2015 CONUS anthropogenic source is 30.6 (29.4-31.3) Tg a$^{-1}$, where the best estimate is from the base inversion and the range is from the inversion ensemble. The 2012 emission total from the EPA GHGI (EPA, 2016) used as prior estimate in our inversion is 28.7 Tg a$^{-1}$, with an uncertainty range 26.4-36.2. We find limited posterior error correlation ($r = 0.33$) between the posterior anthropogenic and natural emission totals. Examining the contributions

from different sectors, our best posterior estimates for landfills and livestock are within 5% of the GHGI, and coal emissions are 6% higher. Oil/gas emissions total 11.1 Tg a$^{-1}$ in our base inversion, 22 (12-32)% higher than the GHGI, and driven by oil/gas production as seen for example in Texas, Oklahoma, and offshore in the Gulf of Mexico. Our national estimates for the emissions from oil and gas production are 3.1 (2.7-3.6) and 5.4 (4.9-5.9) Tg a$^{-1}$, respectively, as compared to 2.3 and 4.4

5   Tg a$^{-1}$ in the GHGI. The posterior error covariance between wetland emissions and both oil production ($r = 0.02$) and gas production ($r = 0.04$) are low, showing that this increase is independent of the large decrease in wetland emissions.

Our scaling factors to the EPA GHGI are for the 2012 emissions as reported by EPA (2016) and used in the inversion as prior estimates. More recently, EPA (2020) updated its methodology for estimating emissions and applied it to a reanalysis

of emissions from previous years including 2012. Changes for 2012 emissions are important for some oil/gas subsectors, as shown in Figure 5. Gas production emissions in 2012 are lower by 19% in the updated GHGI because of a downward correction to emissions from gathering and boosting stations. Oil production emissions in 2012 are 30% lower in the updated EPA GHGI because of previous faulty double-counting of wells. Our correction factor from the inversion increases oil production emissions by a factor 1.9 (1.7-2.3) and natural gas production emissions by a factor 1.5 (1.4-1.6) relative to the updated 2012

GHGI from EPA (2020). The updated GHGI emissions from natural gas processing in 2012 are 55% lower than previously reported, but our inversion finds them to be higher. Our correction factor from the inversion increases gas processing emissions by a factor of 2.9 (2.6-3.1) relative to the updated GHGI.

### 3.2   2010-2015 Emission trends

Figure 6 shows linear emission trends for 2010-2015 optimized by the base inversion (top left panel) and the sensitivity inversion including no interannual variability in prior estimates for wetlands and open fires (top right panel). The base inversion shows a trend of increasing emissions from US wetlands, but this is relative to the prior WetCHARTS estimate of interannual variability of wetland emissions which vary from 31 Tg in 2015 to 36 Tg in 2010. Due to the sparsity of independent constraints, the WetCHARTs inter-annual variations have not been extensively evaluated (Bloom et al., 2017), therefore we have

little confidence in these year-to-year emission changes. The sensitivity inversion including no prior interannual variability for wetlands shows no large trends in US wetland emissions. Both inversions show similar results for the emission trends in anthropogenic source regions. We find an increase in total anthropogenic CONUS emissions of 0.14 Tg a$^{-1}$ a$^{-1}$ (0.4 % a$^{-1}$) over the 2010-2015 period. This anthropogenic trend is much lower than the $2.8 \pm 0.3\%$ a$^{-1}$ increase reported for 2010-2014 by Turner et al. (2016) and more in line with the 2006-2015 trend of $0.7 \pm 0.3$ % a$^{-1}$ in total US emissions estimated by Lan et al. (2019).

The GHGI (EPA, 2020) reports a 0.35 Tg a$^{-1}$a$^{-1}$ decrease in anthropogenic US emissions from 2010 to 2015, at odds with our result. The decrease in the GHGI is mainly driven by decreasing emissions from landfills (-0.10 Tg a$^{-1}$a$^{-1}$) and coal mining (-0.17 Tg a$^{-1}$a$^{-1}$). We find small decreases in the western US that may be related to decreases in emissions from landfills or coal mines (Wyoming). On the national scale, however, the inversion does not detect decreasing emissions from landfills or

**Table 2.** Mean 2010-2015 methane emissions in the contiguous US (CONUS)

| Source (Tg a$^{-1}$) | Prior estimate[a] | Posterior estimate[b] | Sensitivity[c] |
|---|---|---|---|
| Natural | 15.7 | 11.8 (7.1-12.7) | 0.63 |
|    Wetlands | 14.2 | 10.2 (5.6-11.1) | 0.71 |
|    Open fires | 0.5 | 0.4 (0.4-0.5) | 0.13 |
|    Termites | 0.6 | 0.6 (0.6-0.6) | -0.02 |
|    Geological seeps | 0.5 | 0.5 (0.5-0.5) | 0.06 |
| | | | |
| Anthropogenic | 28.7 | 30.6 (29.4-31.3) | 0.53 |
|    Livestock | | | |
|       Enteric Fermentation | 6.7 | 6.9 (6.3-7.0) | 0.16 |
|       Manure Management | 2.5 | 2.5 (2.1-2.5) | 0.22 |
|    Oil and Natural Gas | | | |
|       Gas production | 4.4 | 5.4 (4.9-5.9) | 0.28 |
|       Oil production | 2.3 | 3.1 (2.7-3.6) | 0.53 |
|       Gas Transmission | 1.1 | 1.1 (1.1-1.2) | 0.03 |
|       Gas Processing | 0.9 | 1.1 (1.0-1.2) | 0.41 |
|       Gas Distribution | 0.5 | 0.4 (0.4-0.4) | 0.35 |
|    Landfills | | | |
|       Municipal | 5.2 | 5.0 (4.7-5.0) | 0.26 |
|       Industrial | 0.6 | 0.5 (0.5-0.5) | 0.13 |
|    Coal Mining | | | |
|       Underground | 2.2 | 2.4 (2.3-2.5) | 0.22 |
|       Surface | 0.5 | 0.5 (0.4-0.5) | 0.30 |
|       Abandoned | 0.2 | 0.3 (0.3-0.3) | 0.10 |
|    Wastewater | | | |
|       Municipal | 0.5 | 0.4 (0.4-0.4) | 0.09 |
|       Industrial | 0.2 | 0.2 (0.1-0.2) | 0.16 |
|    Rice Cultivation | 0.5 | 0.4 (0.3-0.5) | 0.28 |
|    Other Anthropogenic[d] | 0.5 | 0.4 (0.4-0.5) | 0.05 |
| | | | |
| Total Source | 44.5 | 42.4 (37.0-42.9) | 0.64 |

[a] The prior estimates include the 2012 EPA GHGI emissions (EPA, 2016) and the 2010-2015 mean of the WetCharts inventory ensemble for wetlands (Bloom et al., 2017).

[b] Posterior estimates from our base inversion, with range from the inversion ensemble in parentheses.

[c] Sensitivity of the posterior estimate to the observations as diagnosed from the averaging kernel matrix, ranging from 0 (no sensitivity, posterior equal to prior) to 1 (full sensitivity, posterior solely determined by the observations). For example, a sensitivity of 0.64 means that 64% of the posterior estimate is constrained by the observations and 36% is constrained by the prior. Averaging kernel sensitivities can statistically be negative in case of error overlap with other sources. The small negative value here is insignificant.

[d] Including fossil fuel combustion, industrial processes, agricultural field burning, and composting.

coal mining.

For total oil/gas emissions we find a US trend of 0.4 (0-1) % $a^{-1}$ for 2010-2015, smaller than the $3.4 \pm 1.4$ % $a^{-1}$ increase reported by Lan et al. (2019) for 2006-2015. The discrepancy may be explained by the different time periods and the fact

that the Lan et al. (2019) oil/gas trend is mainly determined by stations in Oklahoma, North Dakota, and Texas. Most of our increase is driven by the Marcellus Shale area in the Northeast US, amounting to 130 (20-190) Gg $a^{-1}$ $a^{-1}$. This area covering Pennsylvania, Ohio, and West Virginia has seen a large increase in natural gas production driven by unconventional drilling. Natural gas production in the area increased by a factor 7.8 between 2010 and 2015, contributing 22% of US natural gas production in 2015 (EIA, 2020a). If the 130 (20-190) Gg $a^{-1}$ $a^{-1}$ increase is entirely due to gas production, it would only amount

to 15 (2-21)% $a^{-1}$ relative to the mean posterior gas production emissions in the area (0.89 Tg $a^{-1}$) indicating that the leakage rate did decrease over the time period. The latest GHGI shows a national 60 Gg $a^{-1}$ $a^{-1}$ decrease in natural gas production emissions over the 2010-2015 period (EPA, 2020), mainly due to decreasing onshore production and exploration emissions and partly offset by increasing gathering and boosting emissions. For onshore production emissions, the GHGI primarily estimates emissions on the basis of the number of wells, rather than by production rate, and this may underestimate the trend in the

Marcellus Shale as the number of wells only increased by 13% over 2010-2015 (EIA, 2020b) despite the large increase in production. The inversion suggests additional increases over production regions in Texas (Permian Basin) and Oklahoma. The Permian Basin has seen a large increase in production after 2015 (beyond the timespan of our inversion) and is currently the largest oil-producing Basin in the US (Zhang et al., 2020a).

GOSAT provides little information over Canada and Mexico when it comes to trends. There are signs that oil/gas production emissions in both Alberta (Canada) and offshore in the Gulf of Mexico are decreasing, and the latter may be driven by decreasing oil production (Zhang et al., 2019). In Canada, gas production has been stable while oil production from oil sands has increased, but the number of wells has decreased and efforts have been made to reduce emissions (Natural Resources Canada, 2020).

### 3.3 Comparison with other evaluations of the EPA inventory

A number of studies using atmospheric measurements over the US have previously compared their findings to the gridded version of the EPA GHGI reported by Maasakkers et al. (2016) and used as prior estimate in our inversion. Based on an upscaling of facility-level measurements and aircraft data, Alvarez et al. (2018) estimated 2015 US oil/gas emissions of 13 (11-15) Tg

$a^{-1}$, consistent with our posterior estimate of 11.4 (10.3-12.2) Tg $a^{-1}$ for 2015 (posterior mean 2010-2015 emissions plus trend) and much higher than the 7.3 Tg $a^{-1}$ national total from the latest GHGI (EPA, 2020). Similar to our subsector attribution, Alvarez et al. (2018) find the largest difference with the GHGI for the production subsector (factor 2), which they attribute to the GHGI emissions not accounting for emissions from abnormal operating conditions. While Alvarez et al. (2018) did not distinguish between oil and gas production, our results point at a much larger relative discrepancy with the GHGI for

oil production emissions than natural gas production emissions.

In an inversion of data from two tower networks and one aircraft campaign, Cui et al. (2019) found 2014-2016 California methane emissions to be $2.05 \pm 0.26$ Tg $a^{-1}$. Our posterior estimate of 2015 California emissions is 1.6 (0.8-1.7) Tg $a^{-1}$, representing a significant decrease from the prior estimate of 2.3 Tg $a^{-1}$. This is due to our large reduction of mean 2010-2015 emissions in the Los Angeles Basin. This reduction may be overestimated because of the coarseness of model $CO_2$ used in the proxy retrieval, underestimating $CO_2$ over Los Angeles (Turner et al., 2015). Using aircraft measurements, Ren et al. (2019) found 70% higher oil/gas production emissions in the Marcellus Shale (Pennsylvania/West Virginia) in 2015 compared to the 2012 gridded EPA inventory, which they attribute to an increase in production. We find a 2010-2015 increase in emissions for that region, as discussed above, but 2015 emissions are still only 22% higher than the GHGI. Based on surface observations in the Uintah Basin in Utah, Foster et al. (2017) found good agreement with basin-wide emissions from Karion et al. (2013), and found the gridded EPA inventory to be 45% lower after adjusting emissions based on 2015 production data. Most of the emissions in the Uintah Basin are concentrated in one of our grid cells and that cell is optimized individually in our inversion with good constraints (Fig 3), finding 2015 posterior emissions that are 37(12-66)% higher than the prior and no significant 2010-2015 trend.

Based on aircraft data, Plant et al. (2019) found a factor 2 higher anthropogenic urban methane emissions compared to the gridded EPA inventory over five cities on the US East Coast. We find no such difference but evaluating urban emissions along the East Coast is difficult because of overlap with large wetlands emissions that are themselves highly uncertain. It should be possible in principle to separate urban and wetland emissions on the basis of seasonality but we have little confidence in doing so with the GOSAT data because of the need for a seasonal correction to the model - GOSAT mismatch (Section 2.3) and the uncertainty in the seasonality of wetland emissions (Melton et al., 2013; Poulter et al., 2017).

## 4    Conclusions

We have used 2010-2015 methane column data from the GOSAT satellite instrument in a high-resolution inversion of methane emissions and their trends over North America during that period. The inversion for the contiguous US (CONUS) uses as prior estimate a gridded version of the EPA Inventory of U.S. Greenhouse Gas Emissions and Sinks (GHGI), so that results from the inversion are directly relevant for evaluating the GHGI including the contributions from different sectors/subsectors to national methane emissions. We use a 600-member Gaussian mixture model (GMM) as state vector for the inversion that enables us to achieve high resolution ($0.5° \times 0.625°$) in concentrated source regions, and an analytic solution to the Bayesian inverse problem that includes full characterization of information content and facilitates the computation of an ensemble of sensitivity inversions to estimate uncertainty.

We find a best estimate for mean US anthropogenic emissions in 2010-2015 of 30.6 Tg a$^{-1}$ (range of 29.4-31.3 Tg a$^{-1}$ from the inversion ensemble), slightly higher than the EPA GHGI estimate of 28.7 (26.4-36.2) Tg a$^{-1}$. The difference is mainly from oil and gas production, which we find to be higher by 35% (19-59%) and 22% (11-33%) respectively compared to the GHGI. The most recent version of the GHGI EPA (2020) revises emissions from oil/gas production and gas processing emissions downward, opposite from our results. Thus we find that the estimate of emissions from oil production by EPA (2020) is lower than our result by a factor of 2.

Our best estimate of CONUS wetland emissions is 10.2 (5.6-11.1) Tg a$^{-1}$, representing 24% of total CONUS methane emissions. This is lower than the ensemble mean from the WetCHARTS inventory (14.2 Tg a$^{-1}$) used as prior estimate, and is consistent with previous studies pointing to overestimates in US wetland emissions. More work is needed to understand the underlying processes. We find a similar overestimate in wetland emissions over Eastern Canada. We estimate mean 2010-2015 anthropogenic emissions of 4.5 (4.4-4.7)Tg a$^{-1}$ for Canada and 6.1 (5.5-6.3) Tg a$^{-1}$ for Mexico. We find that oil/gas emissions in the IMP (2012) inventory reported to the UNFCCC are too low by 20%, mainly driven by oil production.

We find from the inversion a 2010-2015 increase in US anthropogenic emissions of 0.14 Tg a$^{-1}$ (0.4 % a$^{-1}$), much lower than previous GOSAT-based estimates but at odds with the latest EPA GHGI that reports a 0.35 Tg a$^{-1}$ decrease in emissions over that period. Our increase appears to be largely driven by the rapid growth of unconventional oil/gas production in the eastern US.

*Author contributions.* JDM and DJJ designed the study. JDM performed the analysis. JDM and MPS performed the simulations. JDM, DJJ, MPS, TRS, HN, JXS, YZ, XL, AAB, KWB, and JRW discussed the results. AAB provided the WetCHARTS emissions and supporting data. RJP provided the GOSAT data and supporting guidance. JDM and DJJ wrote the paper, and all authors provided input on the paper for revision before submission.

*Competing interests.* The authors declare that they have no conflict of interest.

*Acknowledgements.* This research was funded by the NASA Carbon Monitoring System (CMS) program. RJP is funded via the UK National Centre for Earth Observation (NCEO grant numbers: nceo020005 and NE/N018079/1). We thank the Japanese Aerospace Exploration Agency, National Institute for Environmental Studies, and the Ministry of Environment for the GOSAT data and their continuous support as part of the Joint Research Agreement. This research used the ALICE High Performance Computing Facility at the University of Leicester for the GOSAT retrievals. Part of this research was carried out at the Jet Propulsion Laboratory, California Institute of Technology, under a

contract with the National Aeronautics and Space Administration.

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

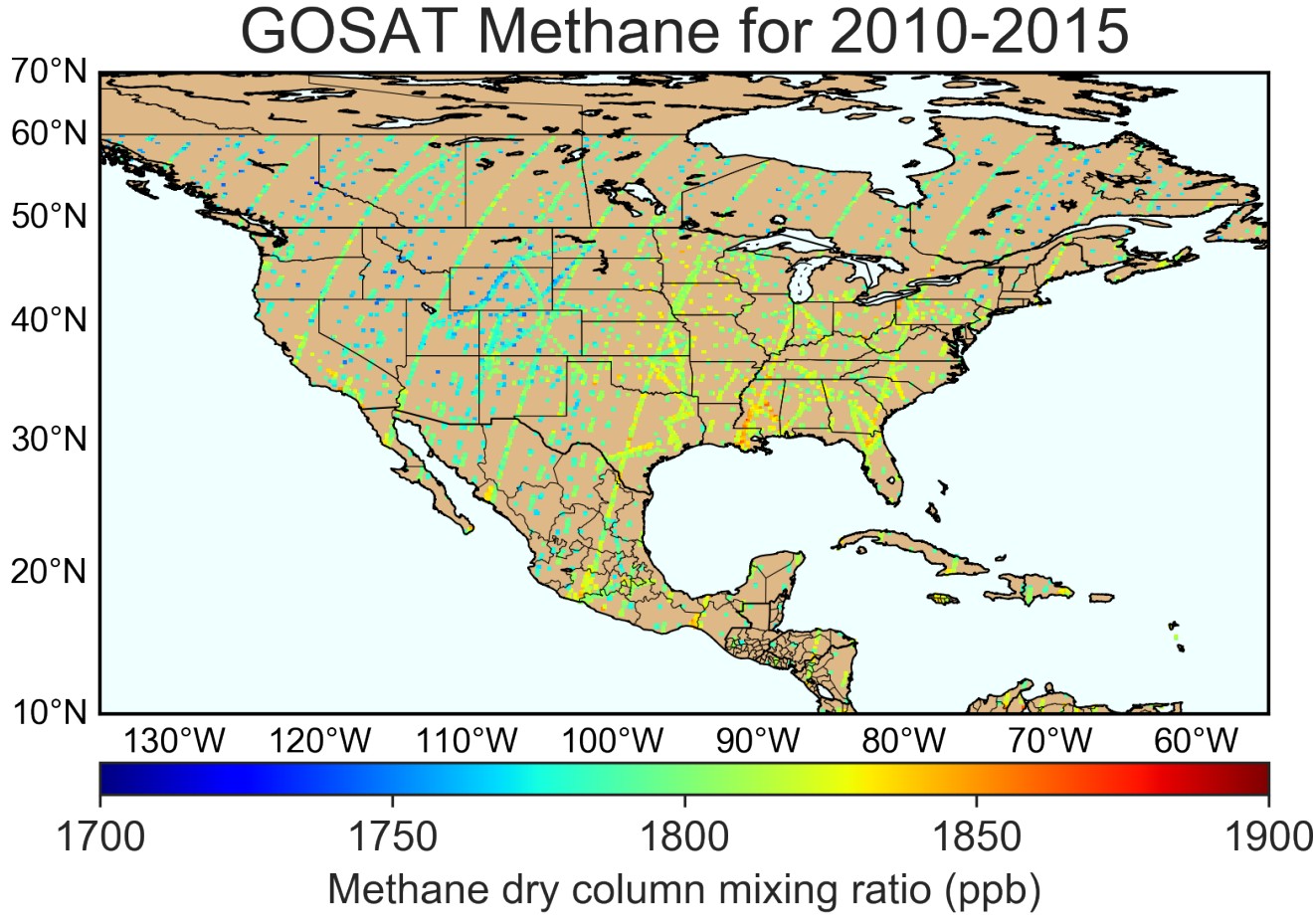

**Figure 1.** Average 2010-2015 methane dry column mixing ratios over North America observed by GOSAT. There are 156110 individual observations over land used in the inversion. The GOSAT data have 10km pixel resolution but we inflate them here to $0.3° \times 0.3°$ for visibility. GOSAT generally takes repeated observations of the same pixels so that most pixels shown here average a number of observations. The apparent north-south tracks that are GOSAT's default mode observations with a repeat cycle of 3 days, while the off-track data are target mode observations. GOSAT observations north of 60N° are excluded because of their seasonal limitation and uncertainty about the stratospheric correction.

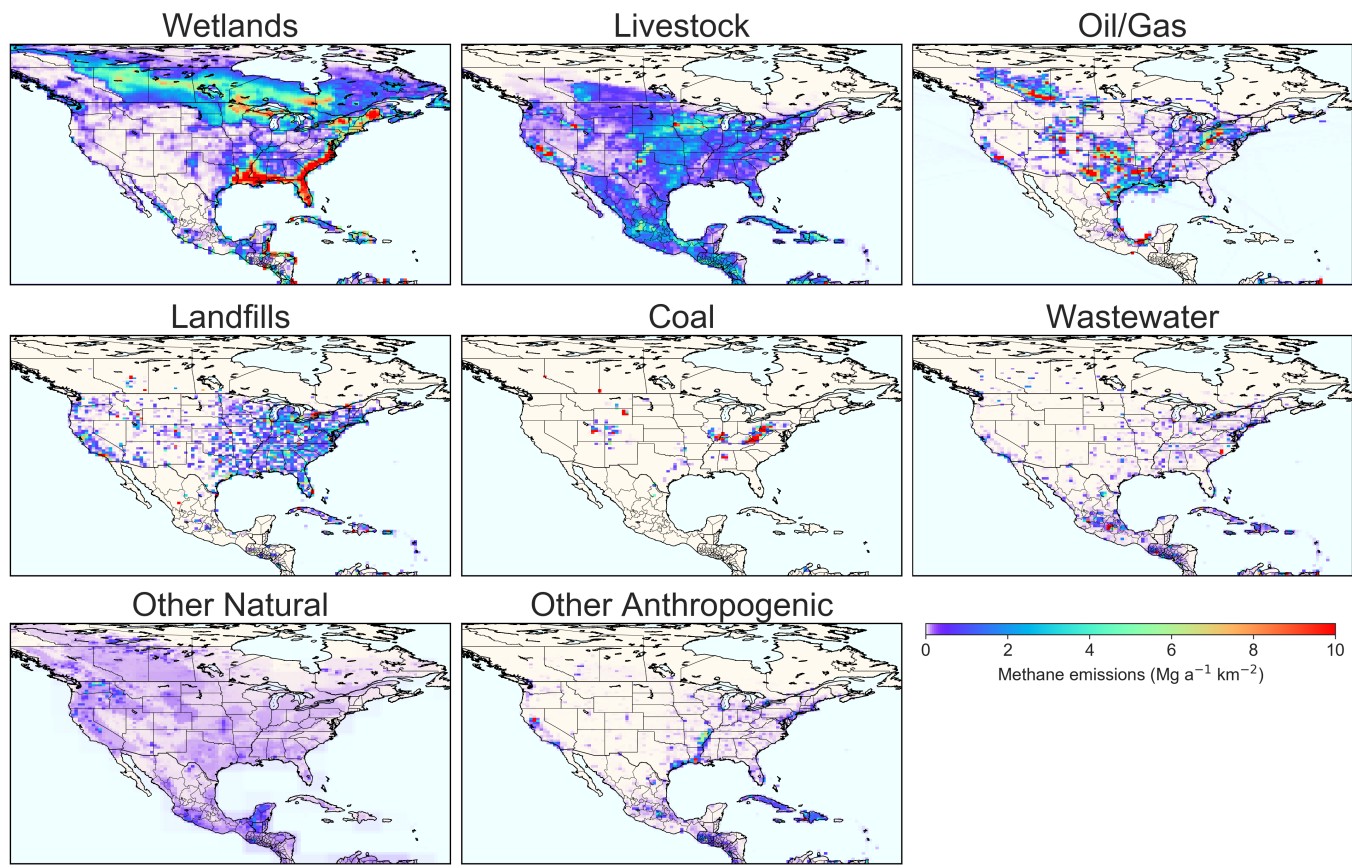

**Figure 2.** Mean prior estimates of methane emissions for 2010-2015. National totals, subsector breakdowns, and references are in Table 1.

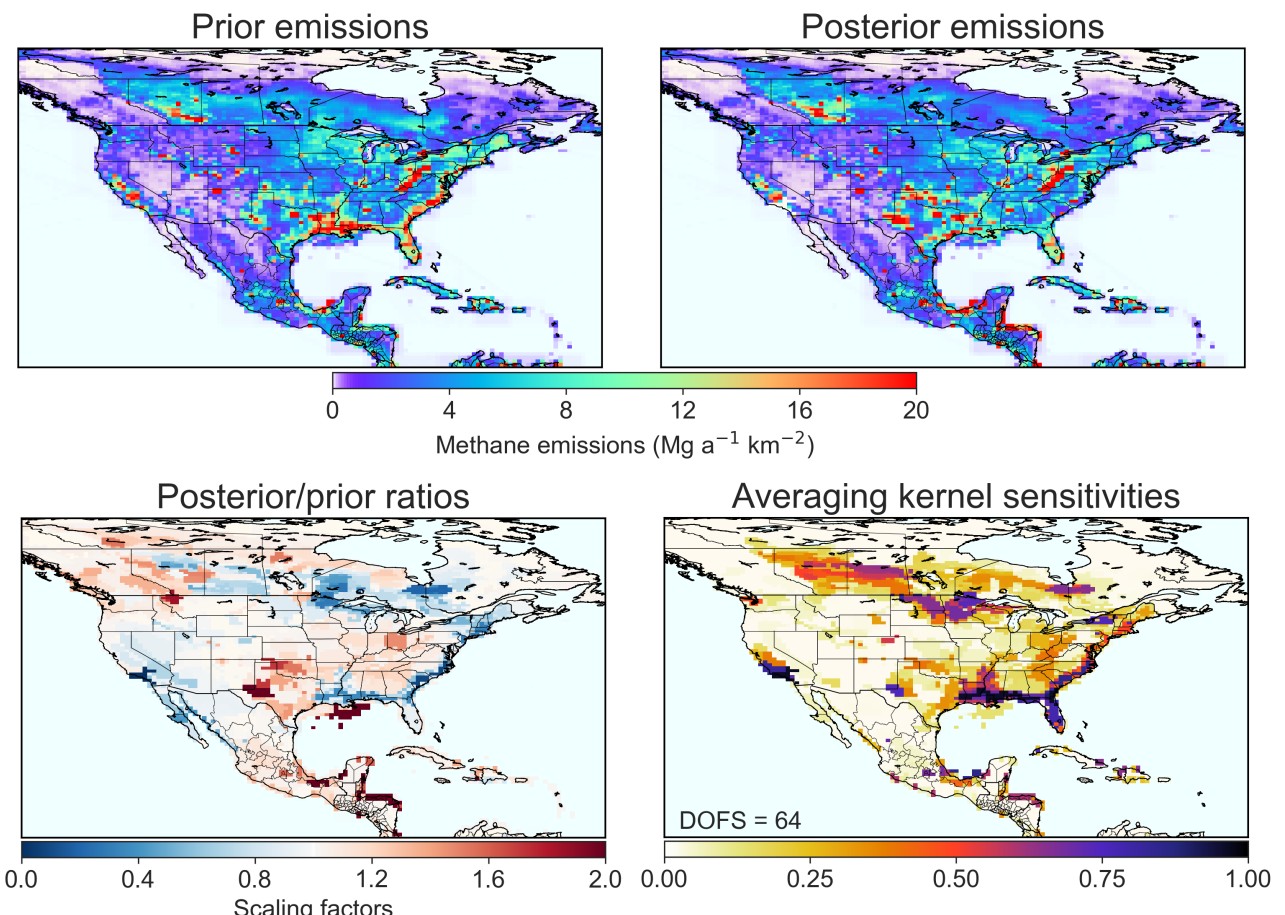

**Figure 3.** Mean 2010-2015 posterior methane emissions projected from the 600-member Gaussian mixture to the $0.5° \times 0.625°$ model grid and comparison to the prior estimate. Results are from the base inversion. The bottom right panel shows the averaging kernel sensitivities projected to the model grid (diagonal elements of the averaging kernel matrix). The trace of the averaging kernel matrix, i.e., the degrees of freedom for signal (DOFS), is given inset. It represents the number of independent pieces of information that can be constrained by the inversion.

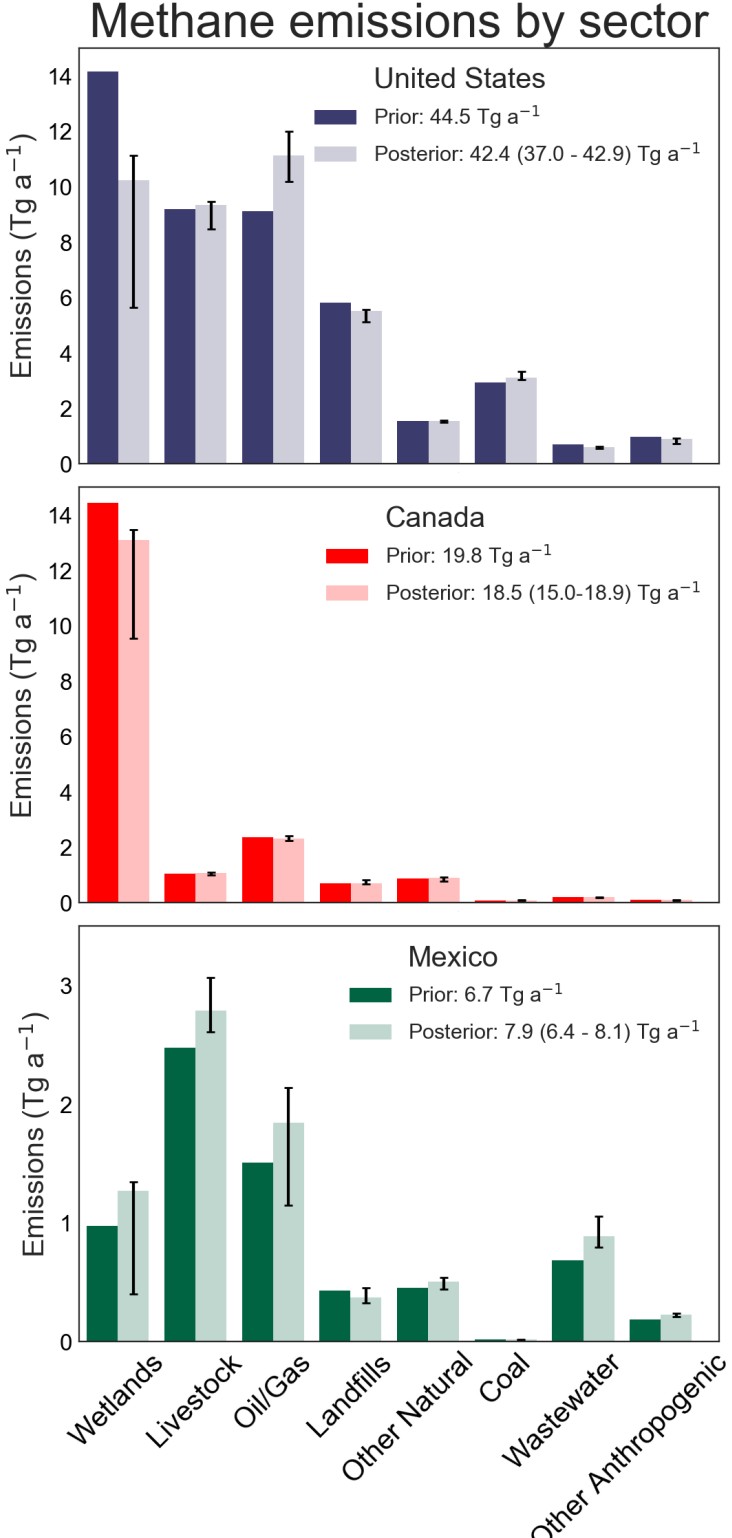

**Figure 4.** Mean 2010-2015 methane emissions per source sector for the contiguous US (CONUS), Canada, and Mexico. Values are shown for the prior estimates (Table 1) and for the posterior estimates after inversion of GOSAT data. Vertical bars show the ranges of results from the inversion ensemble.

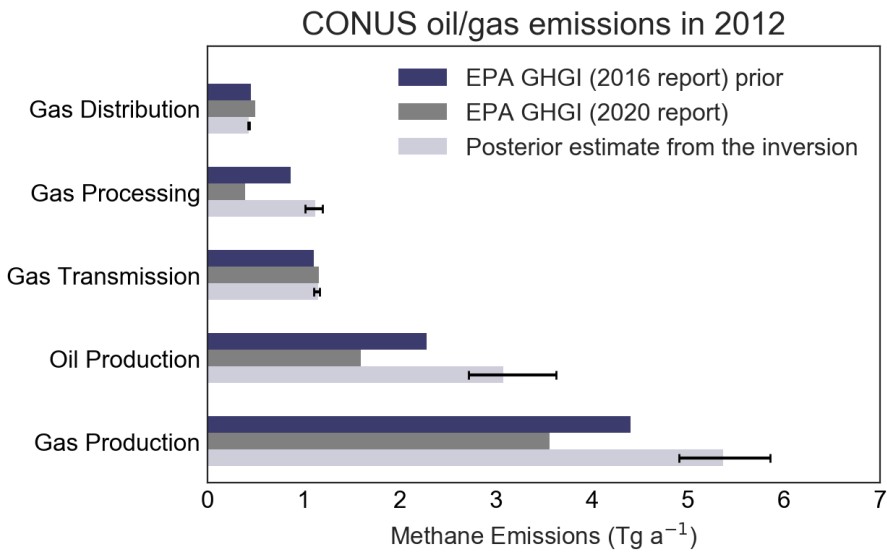

**Figure 5.** Methane emissions from the oil/gas sector in the contiguous US (CONUS) in 20212. The figure shows the original EPA GHGI estimates for 2012 used as prior in the inversion (EPA, 2016), the updated EPA GHGI estimates for 2012 based on revised methodology (EPA, 2020), and the posterior results from the inversion. Horizontal bars give the ranges of the inversion ensemble.

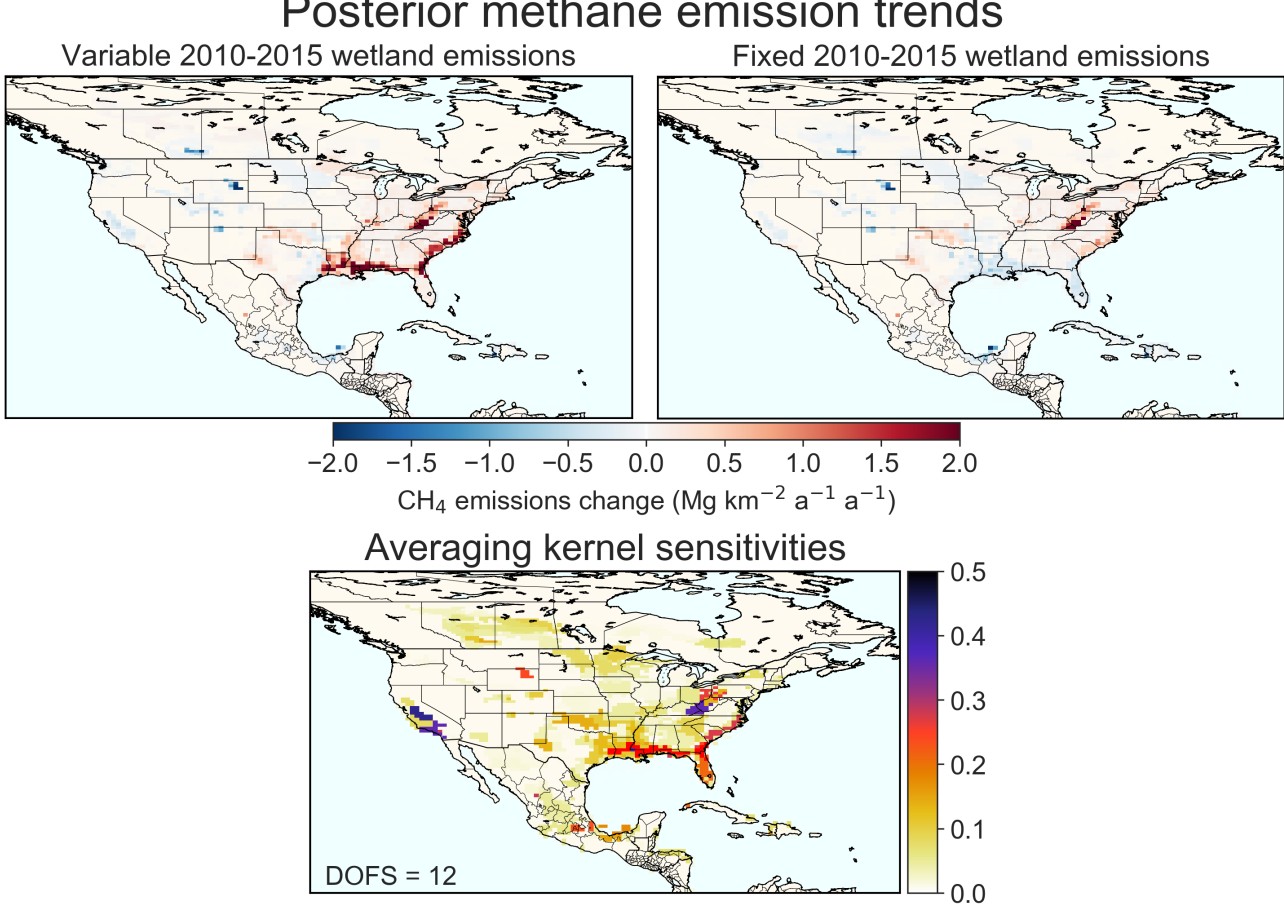

**Figure 6.** 2010-2015 methane emission trends in North America. The top panels show the posterior estimates from the inversion of GOSAT satellite data, allowing for 2010-2015 interannual variability in the prior estimates for wetlands and open fires (base inversion, left) and not allowing for that prior interannual variability (right). The results not allowing for prior interannual variability of wetland and open fire emissions (right) are more reliable. The bottom figure shows the averaging kernel sensitivities, with the degrees of freedom for signal (DOFS) indicated inset.