# Peer review of "2010-2015 North American methane emissions, sectoral contributions, and trends: a high-resolution inversion of GOSAT satellite observations of atmospheric methane"

_Atmospheric Chemistry and Physics, 2020_

## Referee Comment (RC1) · Anonymous Referee #1 · 11 Oct 2020

GENERAL

This is an excellent, well-crafted and important paper. The findings have significant implications for our understanding of US, Canadlan and Mexican methane emissions. They also draw attention to knowledge gaps – for example in quantification of methane fluxes from wetlands. The methodology is detailed and convincingly performed: assuming there are no unexpected errors of computation the results should be sound

and uncertainties appropriate, given the limitations of the data. The work should be published after minor revisions.

SPECIFIC

P2 Line 5 – Very dated flavour to this - maybe add some more up to date work later than Kirschke et al. 2013 - for example Saunois et al papers, and also perhaps recent measurements of global growth. P2 L20 – P3 L7. These paragraphs have a sense of being written a long time ago and only given a quick brush-over update with the addition of brief cursory mentions of the work by Lan et al. 2019 and Bruhwiler et al. 2017. For a non-North American readers (this is a European journal), it would help to give more discussion in these paragraphs to these two recent and important studies. P3 L33 – P4 L1. How much does the cloud cover problem affect GOSAT sensing of wetland emissions? In the equatorial tropics the cloud cover is essentially 100% all through the rainy seasons so it's hard to argue that the cloud-free periods are representative: indeed, they may be warmer. In North America however, with weather system passages, it could be argued that cloud-free episodes are disproportionately in cool clear weather when moisture (and methane) advection is least. Thus GOSAT may disproportionately observe low flux periods. P4 L11 Table 1 – please can this table also estimate the Soil Sink, which is passed over in this Table and not well discussed in the paper as a whole (see also P9 L15). P4 L15-18 – For the priors, the Canadian data are from a business consultancy ICF, and are not from a governmental or peer reviewed source. Thus the use of this data set needs to be defended. Why is this ICF inventory preferred to the Canadian governmental data – does this choice of prior imply that something is assumed to be wrong with the Canadian UNFCCC submission? P4 L5 – Is Fung et al. 1991 the most recent information about termites? P5 L1 – the authors will be aware of the recent publications challenging these estimates of seepage and implying they are much too high. It would be useful to comment on why the higher values are chosen, as they are fairly hard to defend given the ice core evidence. P5 L18 – for those of us not familiar with this correction, maybe add a little justification of

the term – does this reflect the Brewer-Dobson circulation? How much impact on North America is there from 1. the polar vortex downwelling of low methane air, and 2. major convective systems upwelling in the Gulf of Mexico region? P6 Table1 Open fires – 'natural' – though most are human lit. I'm surprised Canada is only 60% of the CONUS: it has some pretty big fires! Seeps number seems too big. Gas distribution in Mexico 0 and production in Mexico 01.? The ICF and IMP data sources are rather old – I wonder if they 'look back' a longish way. Are these priors valid for 2010-15? (NOTE – these comments were written before seeing Fig. 4) P10 L8-9 q10 and CH4:CO2 discussion is interesting and might be worth extending a little as it's important. P11 L108 are significant and the Figure is important and very useful. P12 L8 – 30% lower. Has there also been a reduction in deliberate venting and inefficient flaring? P13 Table 2 is very valuable. P14 L11 – Aha!- at last a brief further comment on Lan et al. It would help to mention Lan et al rather more in this paper. From the European perspective both the Lan and Bruhwiler studies seem to be important and well worth more discussion. P14 L16-17. This is very depressing that landfill emissions are not decreasing. That surely implies a regulatory failure of some magnitude? P14 L18-31 Likewise this paragraph seems to imply that there has been little improvement in practice and technology over the period of study and that all emission changes are due to production shifts (for example to the Marcellus). Is this really so? Have leak and vent reduction efforts and technological improvements really had so little impact? P14 last paragraph and onto p15 – Canada is trying hard to reduce methane emissions – is the emission decrease really only due to production shifts and not to the regulatory and technical changes? Has government action really had no impact? P15 L34 – Separating Urban emissions from wetlands is a global problem as so many cities are in deltas. This is probably beyond what is possible from satellites and needs in situ and aircraft field data - both isotopic sampling and vehicle based mapping.

CONCLUSION. This is an important paper, well written and with significant findings. It should be published after minor revisions

---

## Referee Comment (RC2) · Anonymous Referee #2 · 28 Oct 2020

This paper presents a GEOS-Chem 2010-2015 inversion of $CH_4$ sources over North America. Sectoral emissions and their trends are optimized using a state defined by a so-called Gaussian Mixture Model (GMM), published earlier. Emissions are constrained by GOSAT observations. From an ensemble of inversions, it is found that emissions from the oil and gas sector are higher than in bottom-up reporting. Also, a slight positive trend of 0.4% per year in US anthropogenic methane emissions is derived.

[Figure]

The paper is well written, referencing is adequate, and the results are compared to previous studies. In that respect, the paper is a valuable contribution and deserves publication. However, I also find that the paper cleverly hides some of the bottlenecks in the set-up. I mention two major issues that require further attention/discussion.

First, the inversion is driven by 156110 GOSAT observations in the 2010-2015 time-frame. On page 9, line 29, the authors write that the mean squared difference with GOSAT is reduced by only 3.5% by optimizing the emissions. This relatively small improvement is ascribed to the already good fit using prior emissions, due to the optimized boundary conditions (global Geos-Chem inversion in which emission are optimized by GOSAT observations). This implies that the GOSAT observations are used twice: (1) in the global observations to set the boundary conditions, and (2) in the regional inversion using these boundary conditions. Although I think this is not a major issue, some mention of this drawback is needed (other approaches have been developed to circumvent this issue, e.g. Rödenbeck, C., Gerbig, C., Trusilova, K., and Heimann, M.: A two-step scheme for high-resolution regional atmospheric trace gas inversions based on independent models, Atmos. Chem. Phys., 9, 5331–5342, https://doi.org/10.5194/acp-9-5331-2009, 2009.) Next to this, I was surprised by the large increase in correlation with independent surface observations (r-squared increases from 0.58 to 0.81). Presumably, emissions (and their associated seasonal cycles) are adjusted such that temporal correlations increase. However, the authors provide remarkably little information. In fact, we do not get any information (other than the numbers above) about the ability of the posterior model to simulate GOSAT and surface observations. I also wonder why both data-sources are not assimilated together. Likely there is an unmentioned bias. To remedy this, I suggest that the authors present metrics/figures concerning prior/posterior mismatches with assimilated and unassimilated data.

Second, the authors mention that they developed an analytical optimization, based on 1200 model simulations. This makes it easy to explore sensitivities once the simulations have been performed. In an analytic inversion framework, the calculation of the posterior co-variance matrix is possible. However, on page 11, the authors only present a metric that indicates how well the observations constrain the emissions of particular emissions (actually, I would move this part to the method section). One aspect is missing in this analysis: It would be interesting to know what is the co-variation of total wetland emissions and anthropogenic emissions, because natural emissions (are reduced from 15.7 to 11.8 Tg per year) can only to some extend be separated from anthropogenic emissions (which increase from 28.7 to 30.6 Tg per year). The co-variance matrix would inform on this co-variance (how well can you separate these emissions?), as well as on the uncertainty reduction associated with individual emissions. Although the "sensitivity" inversions provide useful information, I think their range remains always somewhat subjective, depending on the choices made. For instance, is the range in gamma values (0.1, 0.5, 1.0) logical? Why is there no sensitivity for different/perturbed boundary conditions? In that sense, the posterior co-variance of a particular inversion is a useful additional metric that should be reported, when its calculation is feasible (which I guess is the case, given the fact that the averaging kernel matrix is explored).

Finally, I include an annotated pdf which contains minor comments and suggestions.

Please also note the supplement to this comment:
https://acp.copernicus.org/preprints/acp-2020-915/acp-2020-915-RC2-supplement.pdf

**Supplement:**

**2010-2015 North American methane emissions, sectoral contributions, and trends: a high-resolution inversion of GOSAT satellite observations of atmospheric methane**

Joannes D. Maasakkers1,2, Daniel J. Jacob1, Melissa P. Sulprizio1, Tia R. Scarpelli1, Hannah Nesser1, Jianxiong Sheng1,3, Yuzhong Zhang1,4,5,6, Xiao Lu1, A. Anthony Bloom7, Kevin W. Bowman7,8, John R. Worden7, and Robert J. Parker9,10

1Harvard University, Cambridge, Massachusetts 02138, United States

2SRON Netherlands Institute for Space Research, Utrecht, The Netherlands

5School of Engineering, Westlake University, Hangzhou, Zhejiang Province, China

6Institute of Advanced Technology, Westlake Institute for Advanced Study, Hangzhou, Zhejiang Province, China

7Jet Propulsion Laboratory, California Institute of Technology, Pasadena, CA, USA

8Joint Institute for Regional Earth System Science and Engineering, University of California, Los Angeles, CA, USA

9Earth Observation Science, School of Physics and Astronomy, University of Leicester, Leicester, UK

10NERC National Centre for Earth Observation, Leicester, UK

**Correspondence:** J.D. Maasakkers (j.d.maasakkers@sron.nl)

Abstract. We use 2010-2015 GOSAT satellite observations of atmospheric methane columns over North America in a high-resolution inversion of methane emissions, including contributions from different sectors and long-term trends. The inversion involves analytical solution to the Bayesian optimization problem for a Gaussian mixture model (GMM) of the emission field with up to  $0.5^{\circ} \times 0.625^{\circ}$  resolution in concentrated source regions. Analytical solution provides a closed-form characteriza-

- 5 tion of the information content from the inversion and facilitates the construction of a large ensemble of solutions exploring the effect of different uncertainties and assumptions. Prior estimates for the inversion include a gridded version of the EPA Inventory of U.S. Greenhouse Gas Emissions and Sinks (GHGI) and the WetCHARTS model ensemble for wetlands. Our best estimate for mean 2010-2015 US anthropogenic emissions is 30.6 (range: 29.4-31.3) Tg a-1, slightly higher than the gridded EPA inventory (28.7 (26.4-36.2) Tg a-1). The main discrepancy is for the oil and gas production sectors where we find higher
- 10 emissions than the GHGI by 35% and 22% respectively. The most recent version of the EPA GHGI revises downward its estimate of emissions from oil production and we find that these are a factor 2 lower than our estimate. Our best estimate of US wetland emissions is 10.2 (5.6-11.1) Tg a-1, on the low end of the prior WetCHARTS inventory uncertainty range (14.2 (3.3-32.4) Tg a-1) and calling for better understanding of these emissions. We find an increasing trend in US anthropogenic emissions over 2010-2015 of 0.4% a-1, lower than previous GOSAT-based estimates but opposite to the decrease reported
- 15 by the EPA GHGI. Most of this increase appears driven by unconventional oil/gas production in the eastern US. We also find that oil/gas production emissions in Mexico are higher than in the nationally reported inventory, though there is evidence for a

<sup>3Massachusetts Institute of Technology, Cambridge, MA, United States

<sup>4Environmental Defense Fund, Washington, DC, USA

[revised manuscript text omitted]

---

## Author Comment (AC1) · 26 Dec 2020

We thank the reviewers for their useful and knowledgeable comments that have improved our paper. Our responses and manuscript with tracked changes are included in the supplement.

Please also note the supplement to this comment:

[Figure]

https://acp.copernicus.org/preprints/acp-2020-915/acp-2020-915-AC1-supplement.pdf

**[ACPD](https://acp.copernicus.org)**

---

## Author Response (AR1)

We thank both reviewers for their comments.

**Reviewer 1**

**General comments**

This is an excellent, well-crafted and important paper. The findings have significant implications for our understanding of US, Canadlan and Mexican methane emissions. They also draw attention to knowledge gaps – for example in quantification of methane fluxes from wetlands. The methodology is detailed and convincingly performed: assuming there are no unexpected errors of computation the results should be sound and uncertainties appropriate, given the limitations of the data. The work should be published after minor revisions.

**Specific comments**

P2 Line 5 – Very dated flavour to this - maybe add some more up to date work later than Kirschke et al. 2013 - for example Saunois et al papers, and also perhaps recent measurements of global growth.

We have added Saunois et al. (2020) as a reference and described the global growth rate.

*Global methane concentrations have increased by a factor 2.5 compared to preindustrial times Hartmann et al. (2013) and by 7.1 ppb $^{-1}$ since 2007 (with the rate peaking above 10 ppb $a^{-1}$ in 2014 and 2015) after a period of stability in the early 2000s (https: //www.esrl.noaa.gov/gmd/ccgg/trends_ch4/, last access: 20 December 2020).* Major emission source sectors include wetlands (the main natural source), livestock, the fossil fuel industry, and waste management (Kirschke et al., 2013; **Saunois et al., 2020**).

P2 L20 – P3 L7. These paragraphs have a sense of being written a long time ago and only given a quick brush-over update with the addition of brief cursory mentions of the work by Lan et al. 2019 and Bruhwiler et al. 2017. For a non-North American readers (this is a European journal), it would help to give more discussion in these paragraphs to these two recent and important studies.

We added more context on the difference between Turner et al. (2016) and Bruhwiler et al. (2017) and added more information on the Lan et al. (2019) study:

*By contrast, Bruhwiler et al. (2017) found from an ensemble of inversions **using surface and satellite observations** that North American emissions had been flat for 2000-2012 **and that without an inverse model short-term trends can appear to be present in the GOSAT data because of interannual transport variability, choice of background, and the seasonal sampling bias of GOSAT.***

*Using 2006-2015 surface and aircraft observations over the US and Canada, Lan et al. (2019) found a 0.7 ± 0.3 % $a^{-1}$ increase in total US emissions **and a 3.4 ± 1.4 % $a^{-1}$ increase in oil/gas emissions based on stations in Oklahoma, North Dakota, and Texas. They also reported an increase in the ethane:methane emission ratio which could lead to an overestimate of the oil/gas methane trend as inferred from the ethane trend.***

P3 L33 – P4 L1. How much does the cloud cover problem affect GOSAT sensing of wetland emissions? In the equatorial tropics the cloud cover is essentially 100% all through the rainy seasons so it's hard to argue that the cloud-free periods are representative: indeed, they may be warmer. In North America however, with weather system passages, it could be argued that cloud-free episodes are disproportionately in cool clear weather when moisture (and methane) advection is least. Thus GOSAT may disproportionately observe low flux periods.

Through the use of 6 years of data and the ability of the inversion to attribute downwind concentrations to emitting areas with cloud cover during observation, we feel confident the inversion can provide the presented constraints on wetlands. Because of the magnitude of the coastal wetland emissions in our prior, the averaging kernels for these areas are large as later discussed and shown in Figure 3.

P4 L11 Table 1 – please can this table also estimate the Soil Sink, which is passed over in this Table and not well discussed in the paper as a whole (see also P9 L15).

We have added the soil sink as a footnote to the table to be able to add that it is not optimized:

> **The soil sink is 3.6 Tg a$^{-1}$ for the inversion domain (Fung et al., 1991) and is not optimized in the inversion.**

P4 L15-18 – For the priors, the Canadian data are from a business consultancy ICF, and are not from a governmental or peer reviewed source. Thus the use of this data set needs to be defended. Why is this ICF inventory preferred to the Canadian governmental data – does this choice of prior imply that something is assumed to be wrong with the Canadian UNFCCC submission?

We added an explanation for this:

> **The ICF inventory was used as the basis for the Sheng et al. (2017) gridded emissions as it provides a detailed breakdown of sources using methodology similar to the EPA GHGI.**

P4 L5 – Is Fung et al. 1991 the most recent information about termites?

The Fung et al. (1991) termite emissions are in the current standard simulation methane in GEOS-Chem, their global total of 12 Tg a$^{-1}$ fits within the range reported by Saunois et al. (2020) of 9 (3-15) Tg a$^{-1}$. We added the global total to aid in the interpretation.

> *... termite emissions are from Fung et al. (1991)* **with a global total of 12 Tg a$^{-1}$**.

P5 L1 – the authors will be aware of the recent publications challenging these estimates of seepage and implying they are much too high. It would be useful to comment on why the higher values are chosen, as they are fairly hard to defend given the ice core evidence.

The global total of the seepage emissions we use is 5 Tg a$^{-1}$, under the 5.4 Tg a-1 maximum proposed for the pre-indsutrial age by Hmiel et al. (2020). We added the global total and added a comparison to Hmiel et al. (2020) to clarify this.

> *We use geological seepage emissions compiled from literature on both point sources (Etiope, 2015; Kvenvolden and Rogers, 2005) and areal seepage (Kvenvolden and Rogers, 2005; Etiope and Klusman, 2010) as as described in Maasakkers et al. (2019)* **with a global total of 5 Tg a$^{-1}$, under the 5.4 Tg a$^{-1}$ maximum proposed for pre-industrial times by Hmiel et al. (2020) based on ice core measurements**.

P5 L18 – for those of us not familiar with this correction, maybe add a little justification of the term – does this reflect the Brewer-Dobson circulation? How much impact on North America is there from 1. the polar vortex downwelling of low methane air, and 2. major convective systems upwelling in the Gulf of Mexico region?

We added an explanation for the biases and the impact of the polar vortex.

*Following Maasakkers et al. (2019), we correct the GEOS-Chem simulation of GOSAT columns for a latitudinally and seasonally variable background bias likely caused by the extratropical stratosphere (Bader et al., 2016; Saad et al., 2016; Stanevich, 2018).* **The bias is common in atmospheric models and caused by excessive meridional transport in the stratosphere (Patra et al., 2011) and in particular in the seasonal polar vortices (Zhang et al., 2020b).**

P6 Table1 Open fires – 'natural' – though most are human lit. I'm surprised Canada is only 60% of the CONUS: it has some pretty big fires! Seeps number seems too big. Gas distribution in Mexico 0 and production in Mexico 01.? The ICF and IMP data sources are rather old – I wonder if they 'look back' a longish way. Are these priors valid for 2010-15? (NOTE – these comments were written before seeing Fig. 4)

As mentioned with respect to comment P5L1 the global seeps emissions from the database we used are rather small. We have added a footnote to the table to mention all values are rounded and changed all zeros to '<0.1' in the table.

**All values in the table are rounded to one decimal.**

The ICF and IMP comparisons to the UNFCCC data in Section 2.2 are with respect to the latest information available for that time period (UNFCCC, 2020). The years the inventories were based on (2010 and 2013 for Mexico and Canada respectively) are both within the simulation period.

P10 L8-9 q10 and CH4:CO2 discussion is interesting and might be worth extending a little as it's important.

We added some additional discussion:

*The large uncertainty range is driven by the inversion ensemble member without seasonal correction. Based on the root mean square error and spatial correlation, our inversion results are most consistent with the WetCHARTs ensemble members that use GlobCover wetland extent, a q10 = 2 value for the factor increase in the $CH_4$ to $CO_2$ emission ratio per 10 K temperature increase* **(a critical quantity for determining the sensitivity of wetland $CH_4$ production to temperature (Yvon-Durocher et al., 2014; Bloom et al., 2016))**, *and global scaling at the low end or middle of the range (global wetlands emission range 125-166 Tg $a{-}1$).* **A value of q10 = 2 is approximately equivalent with the average $CH_4$ to $CO_2$ temperature sensitivity reported by Yvon-Durocher et al. (2014) based on meta-analyses, which indicates that anaerobic $CH_4$ respiration is substantially more sensitive to temperature relative to overall CO2 respiration rates.**

P11 L108 are significant and the Figure is important and very useful. P12 L8 – 30% lower. Has there also been a reduction in deliberate venting and inefficient flaring?

We have clarified that these changes are not progress over time but rather revisions to the emission factors/data used in the inventory's calculation.

*Our scaling factors to the EPA GHGI are for the 2012 emissions as reported by EPA (2016) and used in the inversion as prior estimates. More recently, EPA (2020) updated its methodology for estimating emissions and applied it to a reanalysis of emissions from previous years including 2012. Changes **for 2012 emissions** are important for some oil/gas subsectors, as shown in Figure 5. Gas production emissions **in 2012** are lower by 19% **in the updated GHGI** because of a downward correction to **emissions from** gathering and boosting **stations**. Oil production emissions **in 2012** are 30% lower **in the updated EPA GHGI because of previous faulty double-counting of wells.** Our correction factor from the inversion increases oil production 10 emissions by a factor 1.9 (1.7-2.3) and natural gas production emissions by a factor 1.5 (1.4-1.6) relative to the updated 2012 GHGI from EPA (2020). **The updated GHGI emissions from natural gas processing in 2012** are 55% lower than previously reported, but our inversion finds them to be higher. Our correction factor from the inversion increases gas processing emissions by a factor of 2.9 (2.6-3.1) relative to the updated GHGI.*

P13 Table 2 is very valuable. P14 L11 – Aha!- at last a brief further comment on Lan et al. It would help to mention Lan et al rather more in this paper. From the European perspective both the Lan and Bruhwiler studies seem to be important and well worth more discussion.

We have extended the discussion of both papers in the introduction as discussed related to comment P2 L20 – P3 L7.

P14 L16-17. This is very depressing that landfill emissions are not decreasing. That surely implies a regulatory failure of some magnitude?

Not necessarily, the change may have been too small or wide-spread and mixed with other emissions to be picked up by the inversion, we rephrased the sentence:

*On the national scale, however, **the inversion does not detect decreasing** emissions from landfills or coal mining.*

Additionally, landfill emissions had already decreased by 31% in 2010 compared to 1990 based on the GHGI.

P14 L18-31 Likewise this paragraph seems to imply that there has been little improvement in practice and technology over the period of study and that all emission changes are due to production shifts (for example to the Marcellus). Is this really so? Have leak and vent reduction efforts and technological improvements really had so little impact?

The increase shown here is much smaller than the relative increase in production over the same area suggesting that the leak rate did in fact decrease, but emissions did increase more than suggested by the EPA GHGI. We added a caveat:

*Most of our increase is driven by the Marcellus Shale area in the Northeast US, amounting to 130 (20-190) Gg $a^{-1}$ $a^{-1}$. This area covering Pennsylvania, Ohio, and West Virginia has seen a large increase in natural gas production driven by unconventional drilling. Natural gas production in the area increased by a factor 7.8 between 2010 and 2015, contributing 22% of US natural gas production in 2015 (EIA, 2020a). **If the 130 (20-190) Gg $a^{-1}$ $a^{-1}$ increase is entirely due to gas production, it would only amount to 15 (2-21)% $a^{-1}$ relative to the mean posterior gas production emissions in the area (0.89 Tg $a^{-1}$) indicating that the leakage rate did decrease over the time period.***

P14 last paragraph and onto p15 – Canada is trying hard to reduce methane emissions – is the emission decrease really only due to production shifts and not to the regulatory and technical changes? Has government action really had no impact?

Because of multiple changes in Canada it is difficult to attribute the apparent decrease that we see, we have added context on Canada in the manuscript:

***In Canada, gas production has been stable while oil production from oil sands has increased, but the number of wells has decreased and efforts have been made to reduce emissions (Natural Resources Canada, 2020).***

P15 L34 – Separating Urban emissions from wetlands is a global problem as so many cities are in deltas. This is probably beyond what is possible from satellites and needs in situ and aircraft field data - both isotopic sampling and vehicle based mapping.

We agree that additional data from either other platforms or newer satellites (TROPOMI) provide exciting prospects to aid the disentanglement of wetland and urban emissions.

CONCLUSION. This is an important paper, well written and with significant findings. It should be published after minor revisions

**Reviewer 2**

**General comments**

This paper presents a GEOS-Chem 2010-2015 inversion of CH4 sources over North America. Sectoral emissions and their trends are optimized using a state defined by a so-called Gaussian Mixture Model (GMM), published earlier. Emissions are constrained by GOSAT observations. From an ensemble of inversions, it is found that emissions from the oil and gas sector are higher than in bottom-up reporting. Also, a slight positive trend of 0.4% per year in US anthropogenic methane emissions is derived. The paper is well written, referencing is adequate, and the results are compared to previous studies. In that respect, the paper is a valuable contribution and deserves publication. However, I also find that the paper cleverly hides some of the bottlenecks in the set-up. I mention two major issues that require further attention/discussion.

First, the inversion is driven by 156110 GOSAT observations in the 2010-2015 timeframe. On page 9, line 29, the authors write that the mean squared difference with GOSAT is reduced by only 3.5% by optimizing the emissions. This relatively small improvement is ascribed to the already good fit using prior emissions, due to the optimized boundary conditions (global Geos-Chem inversion in which emission are optimized by GOSAT observations). This implies that the GOSAT observations are used twice: (1) in the global observations to set the boundary conditions, and (2) in the regional inversion using these boundary conditions. Although I think this is not a major issue, some mention of this drawback is needed (other approaches have been developed to circumvent this issue, e.g. Rödenbeck, C., Gerbig, C., Trusilova, K., and Heimann, M.: A two-step scheme for high-resolution regional atmospheric trace gas inversions based on independent models, Atmos. Chem. Phys., 9, 5331– 5342, https://doi.org/10.5194/acp-9-5331-2009, 2009.)

We have added an additional explanation of the used boundary conditions in section 2.3:

*Three-hourly boundary conditions at the edges of the nested domain are from the 4° × 5° posterior model simulation of Maasakkers et al. (2019), which provides an unbiased fit to the global GOSAT data.* **That posterior simulation includes some information from GOSAT data over the North America domain, which were used (along with the more abundant data outside that domain) in the global inversion; but the main consideration here is to avoid bias in boundary conditions that would otherwise affect the North American inversion.**

Next to this, I was surprised by the large increase in correlation with independent surface observations (r-squared increases from 0.58 to 0.81). Presumably, emissions (and their associated seasonal cycles) are adjusted such that temporal correlations increase. However, the authors provide remarkably little information. In fact, we do not get any information (other than the numbers above) about the ability of the posterior model to simulate GOSAT and surface observations. I also wonder why both data-sources are not assimilated together. Likely there is an unmentioned bias. To remedy this, I suggest that the authors present metrics/figures concerning prior/posterior mismatches with assimilated and unassimilated data.

We have added additional metrics in the manuscript:

*The posterior emissions when implemented in GEOS-Chem reduce the mean squared difference with GOSAT **observations** by 3.5%. **This overall reduction in error** small because random errors in individual observations are large and because the background is already captured well in the prior simulation through the optimized boundary conditions. **The main improvements are found over areas where the averaging kernels are large (Figure 3). For data with averaging kernel sensitivities greater than 0.1, the mean squared difference is reduced by 6.1% and the correlation increases from 0.62 to 0.64.** We independently evaluated the posterior estimate by comparison to in situ methane concentrations from surface sites reported in the GLOBALVIEWplus CH4 ObsPack v1.0 data product compiled by **the** NOAA Global Monitoring Laboratory (Cooperative Global Atmospheric Data Integration Project, 2019). **Compared to the prior simulation (Reduced major axis (RMA) slope: 0.69, $r^2$ = 0.39), the posterior simulation (RMA slope: 0.69, $r^2$ = 0.45) does not degrade the comparison with these data and improves the correlation.** The spatial coefficient of determination between the time-averaged GEOS-Chem and NOAA data increases from $r^2$ = 0.58 with the prior emissions to $r^2$ = 0.81 with the posterior emissions, representing an improvement in our ability to fit observed patterns.*

The NOAA data are not assimilated here to be able to use them as an independent evaluation of the inversion results, combining GOSAT and NOAA is explored in Lu et al. (2020). Here, we have included a comparison between the model and GOSAT on the model resolution for cells with an averaging kernel larger than 0.1 and at least 5 observations below to provide additional context.

[Figure]

Second, the authors mention that they developed an analytical optimization, based on 1200 model simulations. This makes it easy to explore sensitivities once the simulations have been performed. In an analytic inversion framework, the calculation of the posterior co-variance matrix is possible. However, on page 11, the authors only present a metric that indicates how well the observations constrain the emissions of particular emissions (actually, I would move this part to the method section). One aspect is missing in this analysis: It would be interesting to know what is the co-variation of total wetland emissions and anthropogenic emissions, because natural emissions (are reduced from 15.7 to 11.8 Tg per year) can only to some extend be separated from anthropogenic emissions (which increase from 28.7 to 30.6 Tg per year). The co-variance matrix would inform on this co-variance (how well can you separate these emissions?), as well as on

the uncertainty reduction associated with individual emissions. Although the "sensitivity" inversions provide useful information, I think their range remains always somewhat subjective, depending on the choices made. For instance, is the range in gamma values (0.1, 0.5, 1.0) logical? Why is there no sensitivity for different/perturbed boundary conditions? In that sense, the posterior co-variance of a particular inversion is a useful additional metric that should be reported, when its calculation is feasible (which I guess is the case, given the fact that the averaging kernel matrix is explored).

The uncertainties derived from the posterior error covariance matrix for Bayesian inversions are often overly optimistic because they assume fully random errors on the many observations used whereas both model and satellite errors are correlated. This is worse when multiple state vector elements are summed. Even for the results that rely on one Gaussian such as Mexico City, the posterior error is rather small. Based on the posterior error covariance of the base inversion, the uncertainty on the scaling factor would be $1.56 \pm 0.12$, a range much narrower than the 1.56 (1.31-2.20) reported based on the ensemble. We added the reasoning behind this in section 2:

> ***The posterior error covariance matrix from the inversion underestimates the actual uncertainty in the results because of the assumption of fully random observational errors (Jacob et al., 2016). Therefore** we use the range of results from the inversion ensemble as a better measure of uncertainty (Heald et al., 2004).*

The use of the covariance matrix to study the correlation between source sectors is a very interesting suggestion that we implemented in the manuscript, we also added a definition for the mapped posterior error covariance matrix and made an improvement to the calculation of the averaging kernel sensitivities that leads to slightly different values in the table:

> *The summation matrix ($\mathbf{W}$) weighs the relative contribution $w_{i,k}$ of sector/subsector $i$ to the total emission in Gaussian $k$ in the prior inventory:*

$$\mathbf{A_{sub}} = \mathbf{WAW}^*$$
$$\mathbf{\hat{S}_{sub}} = \mathbf{W\hat{S}W^T}$$

> *Here $\mathbf{W}^* = \mathbf{W^T(WW^T)^{-1}}$ is the generalized pseudo-inverse of $\mathbf{W}$, and $\mathbf{A_{sub}}$ (with diagonal elements $a_{i,i}$) and $\mathbf{\hat{S}_{sub}}$ are the averaging kernel matrix and posterior error covariance matrix mapped to the different subsectors. $a_{i,i} = 1$ means that the inversion can fully constrain the national total for that emission category, independent of the prior estimate, while $a_{i,i} = 0$ means that the inversion provides no information and the estimate cannot depart from the prior. **The off-diagonal elements of $\mathbf{\hat{S}_{sub}}$ measure the error correlation in the posterior solution for different subsectors, and this is important to diagnose whether we can optimize different subsectors independently. The diagonal elements of $\mathbf{\hat{S}_{sub}}$ estimate the error variance in the posterior solution for individual subsectors but that estimate is too small because it assumes that the observations are independent and identically distributed (IID condition) (Brasseur and Jacob, 2017). We prefer to estimate the error in the posterior solution from the results of the inversion ensemble, as shown in Table 2.***

> *Our posterior estimate for the mean 2010-2015 CONUS anthropogenic source is 30.6 (29.4-31.3) Tg a$^{-1}$, where the best estimate is from the base inversion and the range is from the inversion ensemble. The 2012 emission total from the EPA GHGI 25 (EPA, 2016) used as prior estimate in our inversion is 28.7 Tg a$^{-1}$, with an uncertainty range 26.4-36.2. **We find limited posterior error correlation (r = 0.33) between the posterior anthropogenic and natural emission totals.** Examining the contributions from different sectors, our best*

*posterior estimates for landfills and livestock are within 5% of the GHGI, and coal emissions are 6% higher. Oil/gas emissions total 11.1 Tg a$^{-1}$ in our base inversion, 22 (12-32)% higher than the GHGI, and driven by oil/gas production as seen for example in Texas, Oklahoma, and offshore in the Gulf of Mexico. Our national estimates for the emissions from oil and gas production are 3.1 (2.7-3.6) and 5.4 (4.9-5.9) Tg a$^{-1}$, respectively, as compared to 2.3 and 4.4 Tg a$^{-1}$ in the GHGI.* ***The posterior error covariance between wetland emissions and both oil production (r = 0.02) and gas production (r = 0.04) are low, showing that this increase is independent of the large decrease in wetland emissions.***

Finally, I include an annotated pdf which contains minor comments and suggestions. These comments are addressed below.

**Specific comments (adopted from PDF comments)**

P1L7 Strange acronym!

The acronym refers to 'Greenhouse Gas Inventory' and is commonly used in the literature, we clarified this in the main text.

P2L1 My first impression is that some words about the BCs are missing.

See response to 'general comments'.

P2L10 What then is the "I". Inventory I guessed?

We clarified the acronym:

*U.S. Greenhouse Gas Emissions and Sinks (**Greenhouse Gas Inventory**, GHGI)*

P2L12 Since we use priors in inversions, this is not totally true.

We rephrased the sentence:

*Measurements of atmospheric methane, including from satellites, can be used through inverse modeling to provide an **evaluation** of these emission estimates (Streets et al., 2013; Jacob et al., 2016).*

P3L3 formatting seems random, e.g. "a" is italic below.

We corrected the italic formatting of 'a' on line 6.

P3L6 Would it be instructive to add a similar picture indicating the amounts of observations. I can imagine that observations get sparser towards 60N. Unclear is the "counter-direction" stripe pattern. Maybe an explanation is needed here?

We clarified the pattern in the observations and their repeat time in the caption of Figure 1:

***The apparent north-south tracks that are GOSAT's default mode observations with a repeat cycle of 3 days, while the off-track data are target mode observations.***

P9L31 This makes me wonder if you are not using the observations multiple times: (1) for the global inversion, setting up the BCs (2) for local constraints.

See response to 'general comments'.

P10L2 Interesting result. Given the fact that the GOSAT improvement is small, this is quite remarkable. Apparently there a re seasonal adjustments driven by GOSAT, that

improve the r^2. But I am also interested in biases. Why are these observations not also assimilated? This sounds that this could be done without any problems. Given the fact that the GOSAT improvement is small, this is a remarkable result.

See response to 'general comments'.

P10L14 But if you use the same observational data, this detail merely comes from better resolution.

We clarified this:

*The large-scale correction patterns revealed by the inversion are similar to those of the coarse (4° × 5°) global inversion reported by Maasakkers et al. (2019), which used the same prior estimates, but we have much more detail here **allowed by the higher resolution of the inversion.***

P10L19 rewrite: i read this sentence a couple of times, without success.

We rephrased the sentence:

***A narrow uncertainty range** does not necessarily reflect confidence in the inversion results. **For small source sectors, it may also be due to insufficient information from the observations so that the optimization is unable to depart from the prior estimate. This can be determined using the averaging kernel sensitivities, as will be done below for the US (see Table 2).***

P11L4 Here I lack information about the Gaussians used. Are there Gaussians centered around Mexico city?

We added this information:

*We also find 56 (31-120) % higher emissions over Mexico City, **which is optimized by a single Gaussian covering five grid cells. The difference is** attributed to wastewater based on the EDGAR spatial patterns. Compared to EDGAR v4.3.2 **a recent gridded inventory for Mexico (Scarpelli et al., 2020)** and the Mexico City Secretariat of Environment (SEDEMA, 2018) air quality emission inventory predict lower emissions from wastewater (68 versus 259 Gg $a^{-1}$ **in the SEDEMA inventory**) but much higher landfill emissions (222 versus 1 Gg $a^{-1}$) indicating that our higher emission estimate may be related to landfill emissions being misallocated in EDGAR v4.3.2.*

P11L23 In theory, this could also be applied to co-variances between sectors. This would be very informative.

See response to 'general comments'.

P11L28 This range is quite subjective because it is determined by your (arbitrary) choice of sensitivity simulations.

See response to 'general comments'.

P13T2 Theoretically, the error reduction in the sum should always be smaller than the individual sub-sectors. I am not sure if this also holds for the ai's.

We added a clarification on the calculation of $a_i$ for the sum of emissions:

> ***We also calculate $a_i$ for the sum of US anthropogenic emission categories and find emissions*** *are 53% informed by the observations, with less information for individual sectors/subsectors.*

> P16L18 It would be interesting to know how wetland (natural) and oil/gas (anthropogenic) emissions are correlated in the posterior. Here it looks like lower wetland emissions go hand-in-hand in higher oil-gas emissions.

See response to 'general comments'.

> P25F3 But these are the Gaussians, right? So add: projected on the model grid.

We have changed the caption:

> *Mean 2010-2015 posterior methane emissions **projected from the 600-member Gaussian mixture to the 0.5º x 0.625º model grid** and comparison to the prior estimate. Results are from the base inversion. The bottom right panel shows the averaging kernel sensitivities **projected to the model grid** (diagonal elements of the averaging kernel matrix).*

We made some other minor textual improvements to improve the flow of the manuscript after incorporation of the feedback from the reviewers.

**Added References**

Bloom, A. A., Exbrayat, J.-F., Van Der Velde, I. R., Feng, L., and Williams, M.: The decadal state of the terrestrial carbon cycle: Global retrievals of terrestrial carbon allocation, pools, and residence times, Proceedings of the National Academy of Sciences, 113, 1285–1290, 2016.

Hartmann, D. L., Tank, A. M. K., Rusticucci, M., Alexander, L. V., Brönnimann, S., Charabi, Y. A. R., Dentener, F. J., Dlugokencky, E. J., Easterling, D. R., Kaplan, A., et al.: Observations: atmosphere and surface, in: Climate Change 2013 the Physical Science Basis: Working Group I Contribution to the Fifth Assessment Report of the Intergovernmental Panel on Climate Change, Cambridge University Press, 2013.

Hmiel, B., Petrenko, V., Dyonisius, M., Buizert, C., Smith, A., Place, P., Harth, C., Beaudette, R., Hua, Q., Yang, B., et al.: Preindustrial $CH_4$ indicates greater anthropogenic fossil $CH_4$ emissions, Nature, 578, 409–412, 2020.

Lu, X., Jacob, D. J., Zhang, Y., Maasakkers, J. D., Sulprizio, M. P., Shen, L., Qu, Z., Scarpelli, T. R., Nesser, H., Yantosca, R. M., Sheng, J., Andrews, A., Parker, R. J., Boech, H., Bloom, A. A., and Ma, S.: Global methane budget and trend, 2010–2017: complementarity of inverse analyses using in situ (GLOBALVIEWplus CH4 ObsPack) and satellite (GOSAT) observations, Atmos. Chem. Phys. Discuss. [preprint], https://doi.org/10.5194/acp-2020-775, in review, 2020.

[revised manuscript text omitted]